# WHERE MATTERS MORE THAN WHAT: DECODING-ALIGNED KV CACHE COMPRESSION VIA POSITION-AWARE PSEUDO-QUERIES

## ABSTRACT

The Key-Value (KV) cache is crucial for efficient Large Language Models (LLMs) inference, but excessively long contexts drastically increase KV cache memory footprint. Existing KV cache compression methods typically rely on input-side attention patterns within a prompt observation window to estimate token importance during the prefill stage. They fail to preserve critical tokens for future generation since these assessments are not derived from the decoding process. Intuitively, an effective observation window should mirror the decoding-stage queries to accurately reflect which tokens the generation process will attend to. However, ground-truth decoding queries are inherently unavailable during inference. For constructing pseudo-queries to approximate them, we find that positional information plays a more critical role than semantic content. Motivated by this insight, we propose decoding-aligned KV cache compression via position-aware pseudo-queries (**DapQ**), a novel and lightweight eviction framework that leverages position-aware pseudo-queries to simulate the output tokens, thereby establishing an effective observation window for importance assessment. It enables precise token eviction that aligns closely with the actual generation context. Extensive evaluation across multiple benchmarks and LLMs demonstrates that DapQ achieves superior performance, particularly under strict memory constraints (e.g., up to nearly lossless performance 99.5% on NIAH with 3% KV cache budgets). Our anonymous code is available at https://anonymous.4open.science/r/Anonymous-DapQ.

## 1 INTRODUCTION

Large Language Models (Achiam et al., 2023; Jiang et al., 2023; Team et al., 2024; Liu et al., 2024a; Grattafiori et al., 2024; Yang et al., 2025a) have achieved significant success across various domains and demonstrated exceptional abilities for processing long-context tasks, such as contextual question answering and document summarization (Liu et al., 2024c; Guo et al., 2024; Liu et al., 2025). A key enabler of efficient inference is the KV cache mechanism, which significantly accelerates autoregressive decoding by reducing the computational complexity of self-attention from $\mathcal{O}(n^2)$ to $\mathcal{O}(n)$. However, with the growth of context length, the memory footprint of the KV cache and the high computational overhead increase dramatically, posing a severe obstacle to the efficient deployment and application of LLMs (Bai et al., 2023).

To tackle these challenges, a wide spectrum of methods has been proposed to compress the KV cache, such as token eviction or merging (Zhang et al., 2023; Tian et al., 2025), quantization (Liu et al., 2024d; Hooper et al., 2024), head or layer-wise sharing (Ainslie et al., 2023; Yang et al., 2024), low-rank decomposition (Dong et al., 2024; Singhania et al., 2024). Among these, token eviction remains a central and widely-adopted strategy. Nevertheless, the rapid growth in input length has further intensified the demand for more effective eviction strategies. In response, as implemented in SnapKV(Li et al., 2024), the observation window has proven superior for retaining critical tokens by combining with pooled accumulated attention scores. This approach is further extended by PyramidKV(Cai et al., 2024), which dynamically allocates layer-wise cache budgets and selects important KV pairs for compression using the window-based attention mechanism. These studies demonstrate the potential of observation windows for effective KV cache compression.

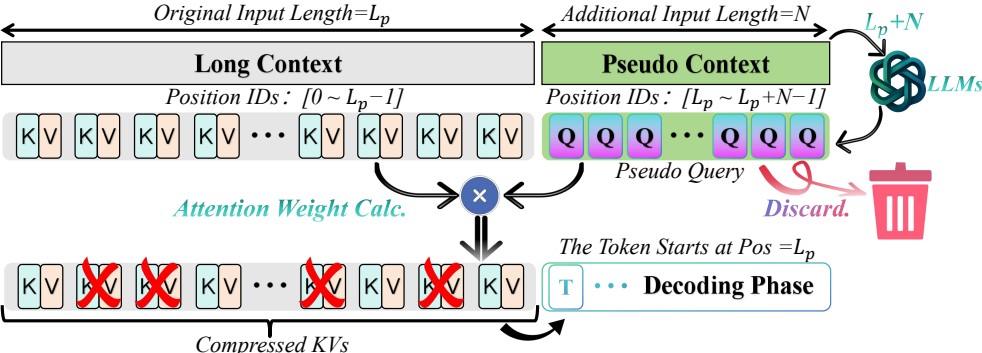

Figure 1: An overview of DapQ. A synthetic pseudo-context (length $N$) is appended to the original input context (length $L_p$), forming an extended sequence of length $L_p+N$. The model processes this sequence during the prefill phase and then obtains pseudo-queries for the synthetic tokens, which are endowed with the correct positional encodings of the first $N$ decoding steps. These pseudo-queries compute attention scores with all keys from the original prompt, establishing the token importance distribution. The $topK$ tokens are retained in the compressed KV cache, while the others, along with all the synthetic tokens are evicted. Autoregressive decoding then begins from the position $L_p$.

However, the input-centric observation window is inherently misaligned with the dynamic query of actual decoding and relies solely on static prompt-based features, typically the last 16-32 tokens. Consequently, they fail to reflect the importance distribution determined by the output-side generation process, leading to a misidentification of the critical tokens for decoding, particularly in complex or noisy contexts. Crucially, ground-truth decoding queries are inherently unavailable during inference, rendering them impractical for directly guiding eviction. To mitigate this, the recent approach LAQ++ (Wang et al., 2025) attempts to better align the observation window with decoding queries by pre-generating pseudo responses, its two-stage eviction process introduces a significant memory peak issue that undermines its practical efficiency. Therefore, constructing effective pseudo-queries to approximate the unavailable future queries without incurring any memory overheads is highly desirable. Inspired by CaliDrop (Su et al., 2025), where queries at adjacent positions exhibit high similarity, our experiments uncover a pivotal insight: **positional information plays a more critical role than semantic content in constructing query approximations and determining attention patterns**. This discovery implies that high-quality pseudo-queries, capable of reliably assessing the importance distribution of KV cache, can be synthesized based on future positional encodings.

Motivated by this insight, we propose decoding-aligned KV cache compression via position-aware pseudo-queries (**DapQ**), a novel and lightweight KV cache eviction framework that constructs pseudo-queries using future positional encodings to accurately simulate the output tokens. These queries serve as an effective observation window for importance scoring, enabling precise cache eviction that aligns closely with the actual generation context. Extensive experiments across multiple benchmarks and four different LLMs demonstrate that DapQ achieves superior performance and outperforms existing eviction baselines, particularly under strict memory constraints.

## 2 RELATED WORK

**Long-Context LLMs.** The growing demand for LLMs to process long contexts intensifies inherent computational and memory challenges. To address these, researchers have proposed various innovations. These include specialized fine-tuning (Chen et al., 2023b) and extending effective context windows through refined positional encoding techniques, such as interpolation and extrapolation (Chen et al., 2023a; Peng & Quesnelle, 2023). Concurrently, to mitigate attention's computational overhead, methods like sparse attention and linear attention have been widely explored (Kitaev et al., 2020; Beltagy et al., 2020; Wang et al., 2020). Beyond traditional Transformer, novel architectures such as State-Space Models (SSMs) (Ye et al., 2025; Gu et al., 2021) offer intrinsically linear complexity solutions for processing extended sequences. Additionally, memory optimization techniques, such as KV cache compression (Xiao et al., 2023; Li et al., 2024), memory offloading(Yang et al., 2025c; Aminabadi et al., 2022) have been developed. And speculative decoding (Leviathan et al.,

2023; Cai et al., 2023) have been proposed to enhance inference throughput. These multifaceted techniques collectively advance LLMs' capabilities in handling ultra-long sequence tasks.

**KV Cache Compression.** KV cache compression is crucial for enhancing the inference efficiency and deployability of LLMs, particularly under resource-constrained scenarios. Various methods have been developed to reduce KV cache memory footprint. Token eviction strategies aim to retain only the most important tokens based on importance metrics like attention scores (Li et al., 2024; Zhang et al., 2023), positional heuristics (Xiao et al., 2023), special tokens (Ge et al., 2023; Chen et al., 2024), or norm-based criteria (Devoto et al., 2024). Complementary to eviction, quantization (Liu et al., 2024d; Hooper et al., 2024) reduces memory by storing less important KV pairs with lower precision, some approaches even achieving sub-2-bit quantization via token-aware and channel-aware techniques. Sharing-based approaches deliver significant memory savings and accelerate inference through head-wise sharing (Shazeer, 2019; Ainslie et al., 2023), inter-layer sharing (Sun et al., 2024; Wu & Tu, 2024; Brandon et al., 2024), or prefix sharing across sequences (Juravsky et al., 2024; Zhu et al., 2024). Low-rank decomposition (Kang et al., 2024; Chang et al., 2024) projects KV cache into lower-dimensional spaces to exploit inherent redundancy, as exemplified by the Multi-Head Latent Attention (Liu et al., 2024b) of DeepSeek, which effectively reduces cache size through low-rank compression and decoupled RoPE while preserving model performance. KV merging(Tian et al., 2025; Cui & Xu, 2025) employs attention-pattern similarity or reparameterization to merge similar semantic information, achieving effectively compression with minimal performance loss. These techniques collectively enable efficient long-context inference.

## 3 OBSERVATION

Given the discussion in Section 1, constructing pseudo-queries to accurately approximate the unavailable ground-truth decoding queries becomes crucial. Building upon CaliDrop's (Su et al., 2025) insight that queries at adjacent positions exhibit high similarity, we hypothesize that this similarity is strongly correlated with positional information rather than semantic content. This prompts us to investigate whether positional information alone can effectively approximate future decoding queries without relying on true decoding content. See Appendix B for details of preliminary experiments.

Table 1: Query similarity comparison under different content and position conditions. This table presents cosine similarities between pseudo-queries and ground-truth decoding queries across four distinct conditions: SC (Same Content), DC (Different Content), SP (Same Position), and DP (Different Position). Post ROPE denotes similarity measured after ROPE has been applied to the query vectors. Pre ROPE indicates similarity measured before ROPE is applied.

| Experiment | Content Similarity | Positional Similarity | Post ROPE | Pre ROPE |
|---|---|---|---|---|
| SC & SP | **Same**("The report discusses the Federal...... Airport Improvement Program (AIP). The program") | **Same**(4424,4425,4426......4453,4454,4455) | 1.0000 | 1.0000 |
| DC & SP | **Different**("Sorry, I don't know. Sorry, I don't know. Sorry, I don't know. Sorry, I don't know. Sorry, I") | **Same**(4424,4425,4426......4453,4454,4455) | 0.7238 | 0.7238 |
| SC & DP | **Same**("The report discusses the Federal...... Airport Improvement Program (AIP). The program") | **Different**(0,1,2......29,30,31) | 0.3267 | 0.7913 |
| DC & DP | **Different**("Sorry, I don't know. Sorry, I don't know. Sorry, I don't know. Sorry, I don't know. Sorry, I") | **Different**(0,1,2......29,30,31) | 0.3522 | 0.7434 |

### 3.1 THE DOMINANCE OF POSITION OVER CONTENT IN QUERY REPRESENTATION

We compare the cosine similarity between various pseudo-queries and the true decoding queries, as detailed in Table 1. Specifically, pseudo-queries assigned the correct future positional IDs but composed of completely irrelevant or nonsensical content (DC&SP), exhibit strong cosine similarity (0.7238) to the actual target decoding queries (SC&SP 1.0000). Conversely, queries with the identical semantic content but incorrect positional IDs (SC&DP vs SC&SP) fail to accurately approximate the target queries (0.3267 vs 1.0000). Notably, the comparison between DC&SP and SC&DP further highlights that maintaining correct positional alignment achieves 2.2× higher similarity than maintaining correct content alone. The stark contrast underscores that the semantic content of queries plays a secondary role compared to their positional encoding in constructing query approximations. The consistently high similarity under Re-ROPE conditions (0.7238 to 0.7913) confirms that the model's underlying processing does not rely heavily on semantic content to distinguish between

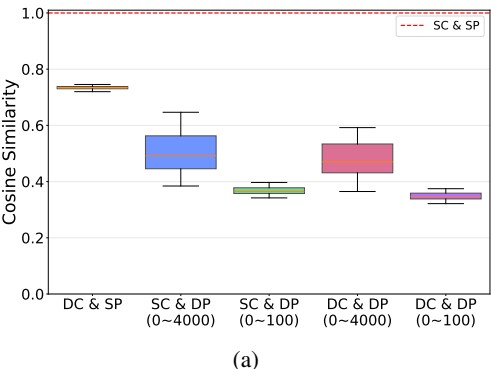 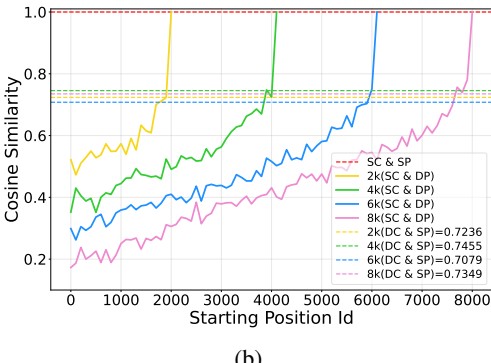

(a)                                                                 (b)

Figure 2: Analysis of Positional Dominance and Offset Sensitivity in Query Similarity. We set the pseudo-queries of fixed length 32. (a) Boxplot of query similarity distributions for a 4k context under different content and position conditions, each aggregated from 100 independent trials. DC: pseudo-queries content is constructed by randomly sampling 32 tokens from the model's vocabulary; DP: pseudo-queries positions are assigned by randomly sampling a consecutive span of 32 index positions from the context length range [0, m] (e.g., [0, 4000]). (b) Query similarity curves over offset positions for contexts of lengths 2k, 4k, 6k, and 8k. The x-axis denotes the starting position assigned to pseudo-queries (e.g., an x-axis value of 3500 corresponds to position IDs $3500 \sim 3531$).

these query vectors, thereby underscoring that the dramatic disparity observed in Post-ROPE scores is almost attributable to the positional information. A large-scale statistical analysis (Figure 2a) confirms the pervasiveness of above phenomenon. From the perspective of query-key interactions, this further implies that the attention mechanism relies heavily on positional information to route information and establish token importance, while exhibiting considerable robustness to variations in semantic content. The results reinforces that deviation from the correct position drastically reduces query similarity, while changes in context have a comparatively minor effect.

## 3.2 THE NECESSITY OF PRECISE POSITIONAL ALIGNMENT

Building upon the dominance of positional information, we further quantify how the positional alignment of pseudo-queries affects their similarity to true decoding queries. As shown in Figure 2b, we fix the pseudo-query content to match the true output and systematically vary their assigned positional IDs. The results reveal a monotonic decay in query similarity **as the absolute offset increases between the assigned position and the correct position**. This decay phenomenon is consistently observed across diverse context lengths (2k to 8k), which is more pronounced in longer context scenarios. The strong sensitivity to positional misalignment underscores that the query approximations depends critically on precise positional information. Consequently, accurately simulating decoding queries requires close alignment with the future generation positional IDs.

## 3.3 POSITION-AWARE PSEUDO-QUERIES FOR TOKEN EVICTION RECALL

The critical question is whether higher query similarity translates into more accurate cache eviction. To quantify this, we evaluate the recall of eviction strategies, which measures its ability to retain the tokens that are most important for the actual generation. Following the methodology of prior work (Wang et al., 2025), the recall rate of the selected KV cache is defined as the proportion of indices selected by the observation window that overlap with those selected by all response tokens from the model. We define the recall metric as follows:

Let $R$ be the set of all tokens in the ground-truth response, and let $K$ be the full key cache from the prefill stage. The gold standard set of indices $M_{\text{gold}}$ for a given budget $B$ is determined by the accumulated attention scores from all true response queries:

$$M_{gold} = \underset{i \in [0, N)}{TopK} \left( \sum_{j \in R} \text{Attention}(q_j, K) \right), \tag{1}$$

where $N$ is the number of all prefill tokens, and $TopK$ returns the indices of the tokens with the highest accumulated scores. The predicted set of indices $M_{\text{pred}}$ is defined analogously to $M_{\text{gold}}$, but

computed using only the queries in a candidate observation window $W$:

$$M_{pred} = \underset{i \in [0,N)}{TopK} \left( \sum_{j \in W} \text{Attention}(q_j, K) \right). \tag{2}$$

The recall is the proportion of the gold-standard tokens that are correctly retained:

$$Recall_W = \frac{|M_{\text{gold}} \cap M_{pred}|}{|M_{gold}|}. \tag{3}$$

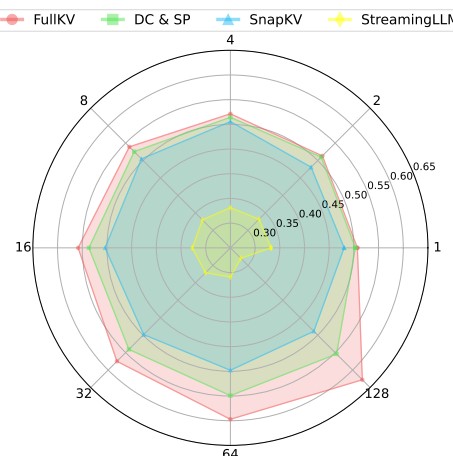

We evaluate this recall metric on the GovReport dataset for different observation windows. As shown in Figure 3, the window composed of pseudo-queries with randomized content but correct future positions achieves significantly better recall than strong baselines like SnapKV and StreamingLLM. Notably, it maintains high recall even as the window size is reduced to 32 or 16, showing the high effectiveness and accuracy of position-based estimation. This demonstrates that a small set of pseudo-queries, informed solely by precise positional forecasting, provides a highly effective basis for importance estimation, enabling accurate eviction even under extreme memory constraints.

In summary, our experiments converge on a pivotal insight: the representation of a query vector is dominated by its positional encoding, with semantic content playing a secondary role. Given that attention

Figure 3: Recall Performance of different methods across various Window Sizes.

scores are derived from query-key interactions, this finding implies that the positional information is the primary determinant in shaping attention patterns. This leads to a profound practical implication: high-fidelity decoding pseudo-queries can be synthesized from positional encodings, entirely bypassing the computationally expensive and memory-intensive process of token generation. This position-aware query approximation forms the foundation of our method.

## 4 METHOD

Motivated by the pivotal insight that positional information dominates query representations, we propose DapQ (as illustrated in Figure 1), a novel KV cache compression framework that accurately simulates decoding-stage contextual positioning during the prefill phase. DapQ synthesizes a decoding-aligned observation window, composed of pseudo-queries endowed with future positional encodings, which mirror the dynamic context of the actual decoding process. This precisely assesses token importance, enabling accurate token eviction without altering the intended timeline.

### 4.1 CONSTRUCTING DECODING-ALIGNED PSEUDO-QUERIES

The core of DapQ is to simulate the dynamic positional query of the decoding phase. For a prompt sequence of length $L_p$, we append a set of $N$ artificially constructed tokens, denoted $\mathbf{T}_{\text{pseudo}}$, to form an extended input sequence. $\mathbf{T}_{\text{pseudo}}$ can be constructed or arbitrarily chosen from the existing context (e.g., uniformly sampled or prefix-suffix concatenation), as their semantic content is secondary to the positional assignment. The crucial operation is to assign $\mathbf{T}_{\text{pseudo}}$ the correct positional indices that they would occupy as the first $N$ tokens generated by the model, rather than arbitrarily:

$$\text{Positions}(\mathbf{T}_{\text{pseudo}}) = [L_p, L_p + 1, ..., L_p + N - 1]. \tag{4}$$

This yields an input sequence with length $L_{\text{total}} = L_p + N$. The model processes this extended sequence during the prefill phase, computing the KV cache for $L_p$ prompt tokens. The primary purpose of this step is to obtain pseudo-queries ($Q_{\text{pseudo}}$) of these $\mathbf{T}_{\text{pseudo}}$, which are endowed with the correct positional encodings for the start of the decoding phase.

## 4.2 Importance Assessment and Token Eviction

We leverage the $Q_{\text{pseudo}}$ to assess the importance of all Keys derived from the original prompt. The importance score for the $j$-th ($j \in [0, L_p - 1]$) prompt token is computed by aggregating its attention scores from each pseudo-query $q_i \in Q_{\text{pseudo}}$:

$$S(j) = \sum_{i=L_p}^{L_p+N-1} \text{Attention}(q_i, k_j). \qquad (5)$$

The $TopK$ tokens with the highest scores $S(j)$ are retained:

$$M_{\text{retain}} = \underset{j \in [0, L_p-1]}{TopK} \left( S(j) \right). \qquad (6)$$

The KV cache is pruned, discarding all key-value pairs not in $M_{\text{retain}}$. **Crucially, the entire synthetic segment $\mathbf{T_{\text{pseudo}}}$ is discarded immediately after performing the importance scoring.** Autoregressive decoding phase then begins from position $L_p$, utilizing only the compressed cache of size $K$. This ensures the model's generation remains consistent with the intended timeline.

## 5 Experiments

### 5.1 Settings

**Models and Benchmarks.** To evaluate the applicability and generalization of DapQ in various models, we conduct experiments on LLaMA-3-8B-Instruct, LLaMA-3.1-8B-Instruct (Grattafiori et al., 2024), Qwen2.5-7B-Instruct (Yang et al., 2025b), and Qwen3-8B (Yang et al., 2025a). To ensure a more comprehensive and robust assessment, we use five benchmarks: LongBench (Bai et al., 2023), LongBenchV2 (Bai et al., 2024), Ruler (Hsieh et al., 2024), HELMET (Yen et al., 2024), and Needle-in-a-Haystack (Kamradt, 2024), each designed to assess distinct aspects of long-context inference, thereby forming a solid foundation for validating DapQ's performance across diverse scenarios.

**Baselines.** To comprehensively validate the performance of DapQ, we select six representative KV cache compression methods as baselines: **FullKV** caches all keys and values for every token, which is the standard approach for KV Cache in transformer-based models; **SnapKV** (Li et al., 2024) captures attention signals from an observation window and employs a clustering algorithm with a pooling layer to select important KV pairs for compression; **PyramidKV** (Cai et al., 2024) leverages cross-layer attention distribution characteristics to dynamically allocate different KV cache budgets and selects important KV pairs for compression; **H2O** (Zhang et al., 2023) identifies Heavy Hitter (H2) tokens based on cumulative attention scores and dynamically balances the retention of recent and H2 tokens to compress KV cache; **StreamingLLM** (Xiao et al., 2023) identifies the attention sink and dynamically balances the retention of recent and initial tokens to compress KV cache; **LaCache** (Shi et al., 2025) adopts a ladder-shaped pattern in the prefilling stage to retain KV of early tokens in shallow layers and gradually shift to later tokens in deeper layers. **Note:** To ensure rigor and consistency, Compression is performed solely during the prefill stage.

**Implementation Details.** For all methods, we set the observation window size to 32 unless otherwise specified (e.g., LaCache use its default settings). In DapQ, the pseudo-queries are constructed by concatenating a small number of tokens from the beginning and the end of the input sequence(e.g., the first 4 and last 28 tokens, the first 2 and last 30 tokens). This design is motivated by two key considerations: the beginning tokens, often high-frequency special tokens (e.g.,`<|begin_of_text|>`), possess stable and generalizable embeddings due to their extensive exposure during training; the ending tokens carry the most recent context, making their semantic state highly relevant to the imminent decoding step. This finding is further supported by Liu et al. (2023). We also validate this strategy through experiments in Fig. 6a, where concatenating prefix and suffix tokens consistently yields superior performance compared to using random or intermediate consecutive tokens from the input as query contents. Complete experiment results are presented in Appendix C.

Table 2: Main Results on LongBench: performance comparison of different KV cache compression methods across 13 datasets on three models (Llama3-8B-Instruct, Qwen2.5-7B-Instruct, Qwen3-8B) under cache sizes of 256, 128, and 64, with complete results available in Appendix C.1.

| | Methods | Single-Document QA | | Multi-Document QA | | Summarization | | Few-shot Learning | | | Synthetic | | Code | | Avg. |
|---|---|---|---|---|---|---|---|---|---|---|---|---|---|---|---|
| | | Qasper | MF-en | HotpotQA | 2WikiMQA | GovReport | MultiNews | TREC | TriviaQA | SAMSum | PCount | PRe | Lcc | RB-P | |
| **Llama3-8B-Instruct** | FullKV | 37.68 | 40.56 | 50.14 | 34.93 | 30.99 | 25.62 | 70.00 | 89.85 | 40.50 | 13.28 | 83.67 | 56.44 | 50.97 | 48.05 |
| | *KV Cache Size = 256* | | | | | | | | | | | | | | |
| | H2O | 28.11 | 36.63 | 48.62 | 31.50 | 21.87 | 21.44 | 45.67 | 89.49 | 38.28 | 12.11 | 83.67 | 61.49 | 53.36 | 44.02 |
| | PyramidKV | 30.88 | 38.11 | 50.20 | 33.88 | 22.54 | 21.84 | 60.00 | 89.26 | 37.07 | 12.78 | 83.67 | 61.34 | 52.51 | 45.70 |
| | SnapKV | 30.84 | 38.39 | 49.75 | 33.80 | 22.18 | 21.53 | 57.00 | 89.65 | 36.97 | 12.11 | 84.00 | 61.78 | 54.92 | 45.61 |
| | DapQ | 32.55 | 38.18 | 50.67 | 34.35 | 22.25 | 21.89 | 60.67 | 90.48 | 38.34 | 11.78 | 83.67 | 62.78 | 55.64 | 46.40 |
| | *KV Cache Size = 128* | | | | | | | | | | | | | | |
| | H2O | 25.95 | 36.25 | 48.65 | 31.90 | 20.79 | 20.30 | 40.00 | 87.29 | 36.25 | 12.33 | 83.67 | 59.81 | 53.14 | 42.79 |
| | PyramidKV | 28.80 | 38.29 | 49.52 | 31.60 | 20.67 | 20.55 | 49.00 | 87.68 | 36.73 | 12.44 | 82.00 | 60.36 | 52.03 | 43.82 |
| | SnapKV | 29.52 | 37.80 | 49.36 | 32.40 | 19.87 | 20.08 | 47.67 | 87.82 | 35.63 | 11.44 | 82.33 | 61.49 | 52.40 | 43.68 |
| | DapQ | 28.76 | 37.24 | 50.04 | 33.59 | 20.47 | 20.63 | 50.00 | 90.06 | 36.87 | 12.11 | 81.67 | 61.81 | 53.92 | 44.40 |
| | *KV Cache Size = 64* | | | | | | | | | | | | | | |
| | H2O | 24.02 | 30.83 | 48.27 | 31.70 | 19.37 | 19.14 | 37.33 | 86.27 | 35.18 | 7.72 | 82.33 | 59.20 | 51.10 | 40.96 |
| | PyramidKV | 22.04 | 31.80 | 47.01 | 31.54 | 15.70 | 16.34 | 39.00 | 76.80 | 32.31 | 10.33 | 79.67 | 55.19 | 47.90 | 38.90 |
| | SnapKV | 25.06 | 32.92 | 47.16 | 31.71 | 16.85 | 17.09 | 40.67 | 86.02 | 33.99 | 11.78 | 78.00 | 57.95 | 50.91 | 40.78 |
| | DapQ | 25.99 | 37.36 | 49.11 | 32.88 | 18.46 | 18.70 | 38.67 | 87.38 | 35.30 | 11.89 | 77.67 | 60.19 | 49.90 | 41.81 |
| **Qwen3-8B** | FullKV | 36.87 | 53.67 | 57.67 | 44.73 | 33.39 | 23.69 | 71.67 | 91.79 | 42.07 | 11.98 | 86.67 | 70.64 | 59.26 | 52.60 |
| | *KV Cache Size = 256* | | | | | | | | | | | | | | |
| | H2O | 27.80 | 44.92 | 51.37 | 40.50 | 22.38 | 18.26 | 46.33 | 90.04 | 38.98 | 12.33 | 86.67 | 66.11 | 53.93 | 46.12 |
| | PyramidKV | 30.41 | 47.81 | 51.00 | 41.37 | 22.36 | 17.41 | 61.67 | 90.96 | 37.65 | 13.00 | 86.33 | 67.01 | 50.11 | 47.47 |
| | SnapKV | 32.40 | 49.29 | 54.38 | 41.40 | 23.79 | 18.99 | 63.00 | 91.07 | 38.57 | 14.67 | 86.67 | 67.99 | 53.77 | 48.92 |
| | DapQ | 32.14 | 50.78 | 54.79 | 44.47 | 24.16 | 19.01 | 62.67 | 91.15 | 39.61 | 14.17 | 86.67 | 67.20 | 53.83 | 49.28 |
| | *KV Cache Size = 128* | | | | | | | | | | | | | | |
| | H2O | 26.60 | 41.37 | 48.10 | 39.85 | 20.83 | 17.27 | 41.33 | 90.21 | 38.16 | 11.72 | 86.67 | 65.22 | 52.77 | 44.22 |
| | PyramidKV | 26.22 | 40.48 | 48.41 | 39.46 | 18.99 | 15.10 | 48.33 | 89.31 | 36.92 | 9.67 | 86.67 | 60.82 | 48.78 | 43.78 |
| | SnapKV | 29.41 | 46.43 | 51.20 | 41.66 | 20.37 | 16.64 | 51.33 | 91.07 | 37.37 | 11.00 | 86.67 | 66.36 | 51.65 | 46.17 |
| | DapQ | 29.10 | 47.15 | 53.84 | 43.00 | 21.11 | 17.27 | 54.00 | 90.45 | 38.50 | 11.67 | 86.00 | 66.29 | 52.03 | 46.95 |
| | *KV Cache Size = 64* | | | | | | | | | | | | | | |
| | H2O | 25.55 | 38.94 | 46.66 | 39.27 | 18.55 | 15.23 | 39.00 | 88.13 | 35.98 | 9.67 | 86.67 | 59.48 | 48.95 | 42.47 |
| | PyramidKV | 25.32 | 40.44 | 46.61 | 39.20 | 16.25 | 12.93 | 44.67 | 88.27 | 34.63 | 11.33 | 83.67 | 59.73 | 46.48 | 42.27 |
| | SnapKV | 25.09 | 39.89 | 46.58 | 39.38 | 15.28 | 12.38 | 42.67 | 87.93 | 35.12 | 11.33 | 84.33 | 57.96 | 46.49 | 41.88 |
| | DapQ | 25.78 | 43.17 | 49.84 | 41.28 | 17.16 | 13.95 | 43.00 | 88.97 | 36.00 | 12.67 | 83.00 | 60.31 | 46.90 | 43.23 |

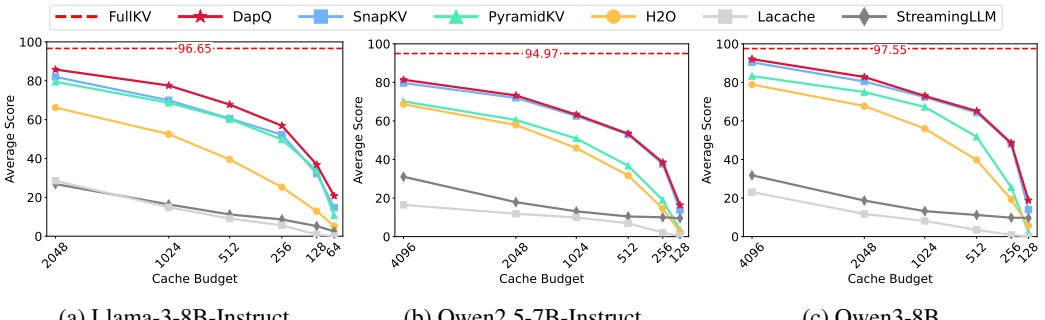

| (a) Llama-3-8B-Instruct | (b) Qwen2.5-7B-Instruct | (c) Qwen3-8B |
|---|---|---|

Figure 4: Average Score on Ruler among 11 datasets across different models.

## 5.2 RESULTS

**LongBench and LongBenchV2 Results.** As shown in Table 2, DapQ outperforms all baselines across every model and cache budget on the LongBench benchmark. The advantage is particularly pronounced under aggressive compression (e.g., budget=64), where DapQ shows a robust ability to retain critical information and mitigate high-compression performance degradation. This superiority is notable on complex reasoning and information integration tasks like HotpotQA and 2WikiMQA, underscoring DapQ's ability in preserving the long-range contextual dependencies and factual knowledge. The more challenging LongBenchV2 benchmark emphasizes deep reasoning over diverse real-world scenarios beyond the mere length of contexts. Under a 64 cache budget, DapQ attains 29.26% accuracy in the category of "Hard", marking a +6.75% absolute improvement over SnapKV (22.51%) on LLaMA3-8B (from Table 4). This strongly confirms that DapQ effectively identifies and retains the critical contextual elements across long contexts by accurately simulating the decoding-stage positional context. The consistent performance advantage across different model architectures shows the strong generalizability and effectiveness of DapQ.

**Ruler and HELMET Results.** We further evaluate DapQ's reasoning capabilities under extreme memory constraints using the RULER benchmark, which tests models on synthetic long-context tasks including multi-hop tracing, retrieval, aggregation, and question answering. Under strict memory constraints, DaqQ exhibits a significant advantage. For instance, Table 5 shows that on the Llama3-8B model with a cache budget of 512, DapQ achieves a notable 59.6% accuracy in the challenging S-NIAH-3 task, substantially outperforming SnapKV (1.4%) and H2O (2.4%). This high-

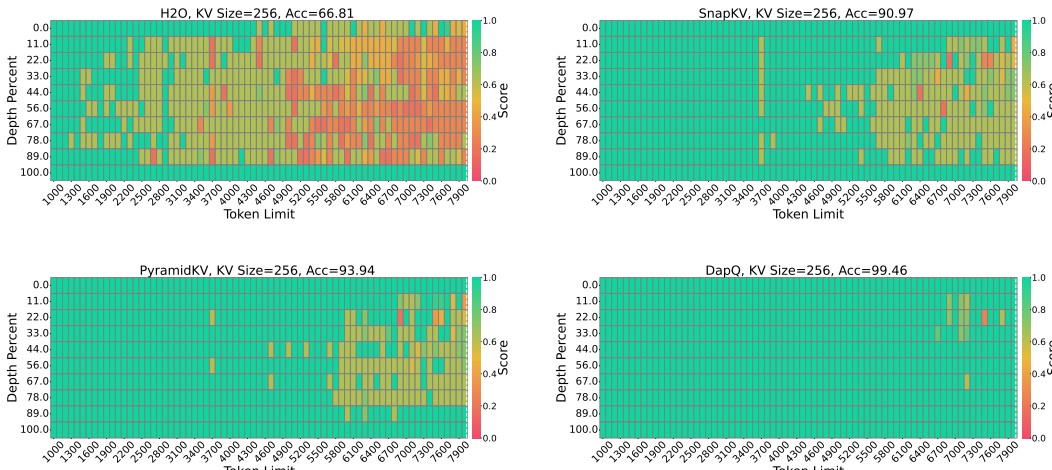

Figure 5: Results of Needle-in-a-Haystack on LLaMA-3-8B-Instruct with 8k context size and 256 KV size. The vertical axis of the figure represents the depth percentage, and the horizontal axis represents the token length.

lights its superior capability in preserving essential information necessary for complex tasks. Furthermore, DapQ consistently demonstrates the highest overall average score across nearly all budget settings on different models, underscoring its robustness and generalizability. Unlike benchmarks focused on synthetic or narrow-domain tasks, HELMET provides a comprehensive, application-centric evaluation across diverse real-world scenarios like retrieval-augmented generation (RAG) and many-shot in-context learning (ICL). On this challenging benchmark, DapQ also demonstrates consistent and superior performance. Table 8 shows that on the Qwen2.5-7B-Instruct with a low cache budget of 512, DapQ achieves an average score of 48.10, outperforming strong baselines SnapKV (43.74), H2O (40.36), and PyramidKV (42.49). Overall, above results further underscore the practical effectiveness and robustness of DapQ in diverse and demanding application scenarios.

**Needle-in-a-Haystack Result.** As shown in Figure 5 and Table 9, DapQ demonstrates universally superior performance compared to all baselines, confirming its consistent effectiveness. A striking example is on LLaMA3-8B with a cache size of 256: DapQ achieves 99.5% accuracy, closely approaching full-cache performance. This exceptional performance shows its ability to maintain key contextual dependencies within constrained memory capacity. DapQ effectively simulates the decoding-stage positional context via prospectively encoded pseudo-queries, enabling precise identification and retention of the key "needles" amidst a vast "haystack" of tokens.

# 6 ANALYSIS

## 6.1 THE IMPACT OF PSEUDO-QUERIES SEMANTIC CONTENT

Our central insight reveals that the semantic content of queries plays a secondary role compared to their positional encoding in determining query representation and attention patterns. To further investigate the practical impact of semantic variation on KV cache compression performance, we conduct an ablation study by evaluating DapQ under a fixed cache budget while altering the semantic content of pseudo-queries. As shown in Figure 6a, the average performance remains highly stable (e.g., coefficient of variation ≈ 1%), regardless of whether the pseudo-query window is constructed from different semantic contents (e.g., the input's prefix and suffix, random in-context tokens, or a fixed nonsensical sequence). This consistency provides compelling empirical support for the conclusion that the attention mechanism relies significantly on positional information to assess token importance, rendering the semantic content a secondary factor.

## 6.2 THE IMPACT OF PSEUDO-QUERIES LENGTH (WINDOW SIZE)

The length of pseudo-queries (the size of the observation window, $N$) is a crucial hyper-parameter, controlling the breadth of the simulated decoding context used for importance estimation. Figure

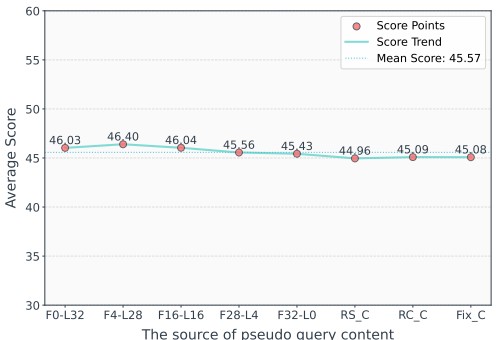 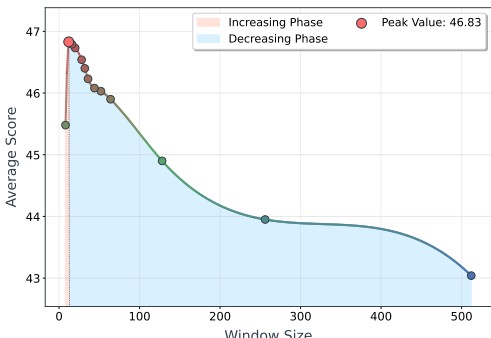

(a) The impact of pseudo-queries quality on Long-Bench performance

(b) The impact of pseudo-queries length on Long-Bench performance

Figure 6: Ablation analysis of pseudo-queries with respect to quality and length. (a) Performance under a fixed pseudo-queries length (i.e., 32), with varying semantic content of pseudo-queries: `Fm_Ln` is the concatenation of the first `m` and the last `n` tokens from the input context; `RS_C` is constructed by concatenating 32 randomly sampled individual tokens from the context; `RC_C` is a randomly sampled consecutive span of 32 tokens from the context; and `Fix_C` is a fixed, repetitive nonsensical sequence (e.g., "Sorry, I don't know. Sorry, I don't know. . . "). (b) Performance under varying observation window sizes $N$, showing a non-monotonic relationship with performance.

6b reveals a non-monotonic relationship between $N$ and performance, characterized by distinct increasing and decreasing phases.

**Increasing Phase (Small $N$):** For small window sizes, performance increases sharply as $N$ grows. This is because a minimal window lacks the contextual breadth to robustly estimate the importance of all relevant tokens, causing high uncertainty in the importance assessment. Adding more pseudo-queries can provide a more comprehensive simulation of the decoding process, leading to a more accurate and holistic importance distribution. This expanded window thereby enables the model to identify and retain a greater number of critical tokens for future effective generation.

**Decreasing Phase (Large $N$):** Beyond a certain point, further increasing $N$ leads to a gradual performance decline. We attribute this to a dilution effect: while the initial queries in the window are precisely aligned with the start of decoding, later queries represent increasingly speculative future positions. The attention patterns for these distant positions become progressively diffuse and attend to tokens less relevant for the initial generation steps, introducing noisier and less reliable signals into the aggregated importance scores. And mutual attention among these queries introduces additional interference, further diverting the focus from critical tokens.

This analysis identifies a "sweet spot" for the window size. The window should be sufficiently sized to capture a representative decoding context but not so large as to dilute the attentional signal. The existence of this optimum confirms that our method is not relying on a brute-force approach. Instead, it performs a precise and efficient simulation by concentrating on the most relevant segment of the decoding trajectory, thereby striking a balance between content fidelity and contextual breadth.

## 7 CONCLUSION

In this work, we introduce DapQ, a novel and effective KV cache compression framework that leverages position-aware pseudo-queries to simulate the output tokens, thereby establishing an effective observation window for importance assessment. During the prefill stage, it enables precise token eviction that aligns closely with the actual generation context. Extensive experiments demonstrate that DapQ consistently outperforms existing baselines and achieves superior performance in long-context scenarios, particularly under strict memory constraints. This work underscores the primacy of positional information over semantic content in constructing query approximations and determining attention patterns. This insight promotes us rethinking of the role of positional context in LLMs optimization and efficient inference. Extending this positional simulation approach to dynamic, layer-wise budget allocation and integrating it with quantization techniques present promising directions for achieving more effective compression.

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

## A    THE USE OF LARGE LANGUAGE MODELS

We use the large language model to aid or polish writing of this paper. This help is purely editorial and do not involve any other contributions.

## B    PRELIMINARY EXPERIMENT DETAILS

### B.1    QUERY SIMILARITY COMPARISON UNDER DIFFERENT CONTENT AND POSITION CONDITIONS.

We conduct the query similarity analysis using a representative example from the GovReport dataset, which is designed for document summarization tasks. The input sequence consists of 4424 tokens. The ground-truth decoding queries are obtained by extracting the first 32 output tokens generated by LLaMA-3-8B-Instruct for this input. The semantic content of these output tokens is: "The report discusses the Federal Aviation Administration's (FAA) state block grant pilot program, which is part of its Airport Improvement Program (AIP). The program", and they are assigned the positional indices 4424-4455. We then construct a pseudo-context of length 32 with the repetitive content: "Sorry, I don't know. Sorry, I don't know. Sorry, I don't know. Sorry, I don't know. Sorry, I". Notably, the positional IDs for the pseudo-context can be flexibly configured to emulate various decoding scenarios.

#### B.1.1    EXPERIMENTAL DETAILS OF QUERY SIMILARITY COMPARISON

In Table 1, we evaluate pseudo-queries by varying two key attributes relative to the ground-truth decoding queries: semantic content and positional assignment. The content is either **consistent with** the true output (i.e., the actual beginning of the model's summary, "The report discusses...") or **different from** it (a fixed, nonsensical sequence, e.g., "Sorry, I don't know..."). Similarly, the positional indices are either **aligned with** the true future decoding positions (4424-4455) or **deviated from** them (assigned to a random consecutive span, e.g., 0-31). The cosine similarity between each set of pseudo-queries and the ground-truth queries is reported under two measurement conditions: *Post ROPE*, which captures the final query representation after the application of Rotary Position Embedding (ROPE), and *Re ROPE*, which reflects the similarity before the positional encoding is applied, thus isolating the effect of semantic content.

#### B.1.2    EXPERIMENTAL DETAILS OF QUERY SIMILARITY DISTRIBUTION ANALYSIS

To quantitatively assess the impact of positional and content variations on query representation, we conduct a large-scale statistical analysis as depicted in Figure 6a. Each box in the boxplots is aggregated from 100 independent trials, providing a stable estimate of the similarity distribution. The **Different Content (DC)** condition is implemented by randomly sampling 32 tokens from the model's full vocabulary, effectively removing any meaningful semantic correlation with the true output. The **Different Position (DP)** condition is implemented by assigning a consecutive span of 32 positions randomly sampled from two distinct ranges to introduce positional deviation: a general deviation range (0-4000), which represents a random mismatch within the context window, and an extreme deviation range (0-100), which is specifically chosen to maximize the absolute offset from the correct positions (i.e., 4424-4455), thereby rigorously testing the hypothesis that positional accuracy is dominant.

### B.2    EXPERIMENTAL DETAILS OF POSITIONAL OFFSET SENSITIVITY

To systematically quantify the sensitivity of query representations to positional miscalibration, we conduct the analysis presented in Figure 2b. The experiment investigates how the similarity between pseudo-queries and the true decoding decays based on the absolute offset between their assigned positional indices and the correct future positions. For this purpose, we select input examples of varying context lengths (2k, 4k, 6k, and 8k tokens) from the GovReport dataset. For each context length, we construct pseudo-queries with fixed semantic content (aligned with the true output) but systematically vary their assigned starting position. The x-axis represents this starting position assigned to pseudo-queries (e.g., an x-axis value of 3500 indicates that the 32 pseudo-queries are

assigned the consecutive position IDs from 3500 to 3531). The y-axis measures the resulting cosine similarity between the pseudo-queries and the ground-truth decoding queries. This approach allows us to observe the monotonic decay in similarity with increasing positional offset.

# C    COMPLETE EXPERIMENT RESULTS AND DETAILS

## C.1    RESULTS AND DETAILS ON LONGBENCH

We comprehensively evaluate the performance of DapQ and baselines on LongBench benchmark with the following setup:

- **Models:** LLaMA-3-8B-Instruct, Qwen2.5-7B-Instruct, Qwen3-8B;
- **KV Cache Budgets:** 256, 128, 64 tokens.

The complete results are shown in Table 3.

## C.2    RESULTS AND DETAILS ON LONGBENCHV2

We comprehensively evaluate the performance of DapQ and baselines on LongBenchV2 benchmark with the following setup:

- **Models:** LLaMA-3-8B-Instruct, LLaMA-3.1-8B-Instruct, Qwen2.5-7B-Instruct, Qwen3-8B;
- **KV Cache Budgets:** 128, 64 tokens.

The complete results are shown in Table 4.

## C.3    RESULTS AND DETAILS ON RULER

We comprehensively evaluate the performance of DapQ and baselines on Ruler benchmark with the following setup:

- **Models:** LLaMA-3-8B-Instruct, Qwen2.5-7B-Instruct, Qwen3-8B;
- **KV Cache Budgets:** 4096, 2048, 1024, 512, 256, 128, 64 tokens.

The complete results are shown in Table 5, Table 6, Table 7.

## C.4    RESULTS AND DETAILS ON HELMET

We comprehensively evaluate the performance of DapQ and baselines on HELMET benchmark with the following setup:

- **Models:** LLaMA-3-8B-Instruct, Qwen2.5-7B-Instruct;
- **KV Cache Budgets:** 2048, 1024, 512, 256, 128 tokens.

The complete results are shown in Table 8.

## C.5    RESULTS AND DETAILS ON NEEDLE-IN-A-HAYSTACK

We comprehensively evaluate the performance of DapQ and baselines on Needle-in-a-Haystack benchmark with the following setup:

- **Models:** LLaMA-3-8B-Instruct, LLaMA-3.1-8B-Instruct, Qwen2.5-7B-Instruct, Qwen3-8B;
- **KV Cache Budgets:** 256, 128, 64 tokens.

The complete results are shown in Table 9.

Table 3: Performance comparison of different methods across various LLMs on LongBench.

| | Methods | Single-Document QA | | Multi-Document QA | | Summarization | | Few-shot Learning | | | Synthetic | | Code | | Avg. |
|---|---|---|---|---|---|---|---|---|---|---|---|---|---|---|---|
| | | Qasper | MF-en | HotpotQA | 2WikiMQA | GovReport | MultiNews | TREC | TriviaQA | SAMSum | PCount | PRe | Lcc | RB-P | |
| **Llama3-8B-Instruct** | FullKV | 37.68 | 40.56 | 50.14 | 34.93 | 30.99 | 25.62 | 70.00 | 89.85 | 40.50 | 13.28 | 83.67 | 56.44 | 50.97 | 48.05 |
| | *KV Cache Size = 256* | | | | | | | | | | | | | | |
| | H2O | 28.11 | 36.63 | 48.62 | 31.50 | 21.87 | 21.44 | 45.67 | 89.49 | 38.28 | 12.11 | 83.67 | 61.49 | 53.36 | 44.02 |
| | PyramidKV | 30.88 | 38.11 | 50.20 | 33.88 | **22.54** | 21.84 | 60.00 | 89.26 | 37.07 | **12.78** | 83.67 | 61.34 | 52.51 | 45.70 |
| | SnapKV | 30.84 | **38.39** | 49.75 | 33.80 | 22.18 | 21.53 | 57.00 | 89.65 | 36.97 | 12.11 | **84.00** | 61.78 | 54.92 | 45.61 |
| | DapQ | **32.55** | 38.18 | **50.67** | **34.35** | 22.25 | **21.89** | **60.67** | **90.48** | **38.34** | 11.78 | 83.67 | **62.78** | **55.64** | **46.40** |
| | *KV Cache Size = 128* | | | | | | | | | | | | | | |
| | H2O | 25.95 | 36.25 | 48.65 | 31.90 | **20.79** | 20.30 | 40.00 | 87.29 | 36.25 | 12.33 | **83.67** | 59.81 | 53.14 | 42.79 |
| | PyramidKV | 28.80 | **38.29** | 49.52 | 31.60 | 20.67 | 20.55 | 49.00 | 87.68 | 36.73 | **12.44** | 82.00 | 60.36 | 52.03 | 43.82 |
| | SnapKV | 29.52 | 37.80 | 49.36 | 32.40 | 19.87 | 20.08 | 47.67 | 87.82 | 35.63 | 11.44 | 82.33 | 61.49 | 52.40 | 43.68 |
| | DapQ | 28.76 | 37.24 | **50.04** | **33.59** | 20.47 | **20.63** | **50.00** | **90.06** | **36.87** | 12.11 | 81.67 | **61.81** | **53.92** | **44.40** |
| | *KV Cache Size = 64* | | | | | | | | | | | | | | |
| | H2O | 24.02 | 30.83 | 48.27 | 31.70 | **19.37** | **19.14** | 37.33 | 86.27 | 35.18 | 7.72 | **82.33** | 59.20 | **51.10** | 40.96 |
| | PyramidKV | 22.04 | 31.80 | 47.01 | 31.54 | 15.70 | 16.34 | 39.00 | 76.80 | 32.31 | 10.33 | 79.67 | 55.19 | 47.90 | 38.90 |
| | SnapKV | 25.06 | 32.92 | 47.16 | 31.71 | 16.85 | 17.09 | **40.67** | 86.02 | 33.99 | 11.78 | 78.00 | 57.95 | 50.91 | 40.78 |
| | DapQ | **25.99** | **37.36** | **49.11** | **32.88** | 18.46 | 18.70 | 38.67 | **87.38** | 35.30 | **11.89** | 77.67 | **60.19** | 49.90 | **41.81** |
| **Qwen2.5-7B-Instruct** | FullKV | 36.50 | 49.70 | 55.91 | 44.70 | 31.64 | 22.84 | 66.33 | 89.34 | 42.49 | 11.00 | 86.33 | 61.97 | 59.97 | 50.67 |
| | *KV Cache Size = 256* | | | | | | | | | | | | | | |
| | H2O | 27.76 | 42.94 | 47.89 | 40.97 | 22.13 | 18.31 | 44.33 | 84.51 | 39.77 | 10.67 | 86.00 | 57.07 | 54.09 | 44.31 |
| | PyramidKV | 29.65 | 46.37 | 49.89 | 40.77 | 20.42 | 16.80 | 53.00 | 87.89 | 39.61 | 10.67 | 86.00 | 52.61 | 49.49 | 44.82 |
| | SnapKV | **31.12** | **47.87** | 51.76 | 40.78 | 22.25 | **18.41** | 53.33 | 87.66 | 39.34 | 10.67 | 86.00 | 56.87 | **54.36** | 46.19 |
| | DapQ | 30.21 | 45.00 | **51.92** | **41.46** | **22.35** | 18.40 | **56.67** | **88.64** | **39.92** | **11.00** | 86.00 | **58.40** | 53.82 | **46.45** |
| | *KV Cache Size = 128* | | | | | | | | | | | | | | |
| | H2O | 26.83 | 37.80 | 45.14 | 39.77 | **20.13** | 16.64 | 40.67 | 81.43 | 38.56 | 10.67 | **85.67** | 53.97 | 51.52 | 42.22 |
| | PyramidKV | 26.37 | 43.09 | 46.57 | 39.07 | 18.01 | 15.15 | 43.33 | 84.69 | 38.40 | 10.67 | 84.67 | 49.45 | 46.90 | 42.03 |
| | SnapKV | **27.46** | 41.81 | 48.62 | 41.40 | 19.42 | 16.19 | 42.33 | 83.91 | **38.89** | 10.67 | 85.00 | 52.14 | 51.39 | 43.02 |
| | DapQ | 26.81 | **43.12** | **49.62** | 41.36 | 19.89 | **16.68** | **47.00** | **84.92** | 38.07 | **11.00** | 85.00 | **54.46** | **51.70** | **43.82** |
| | *KV Cache Size = 64* | | | | | | | | | | | | | | |
| | H2O | 24.33 | 32.36 | 44.94 | 39.06 | **17.96** | **15.05** | 37.33 | 82.76 | 35.27 | 10.67 | **85.00** | 49.44 | 45.52 | 39.98 |
| | PyramidKV | 22.36 | 35.19 | 43.51 | 38.08 | 14.04 | 11.10 | 37.33 | 84.81 | 35.58 | 10.67 | 80.33 | 44.67 | 42.22 | 38.45 |
| | SnapKV | 22.90 | 40.66 | 45.56 | 40.28 | 15.42 | 12.03 | 37.67 | **85.11** | 36.92 | 10.67 | 82.00 | 46.16 | 45.75 | 40.09 |
| | DapQ | **25.58** | **42.65** | **49.66** | **41.25** | 16.90 | 14.05 | **39.67** | 84.16 | 35.49 | **11.00** | 80.67 | **49.38** | **46.77** | **41.33** |
| **Qwen3-8B** | FullKV | 36.87 | 53.67 | 57.67 | 44.73 | 33.39 | 23.69 | 71.67 | 91.79 | 42.07 | 11.98 | 86.67 | 70.64 | 59.26 | 52.60 |
| | *KV Cache Size = 256* | | | | | | | | | | | | | | |
| | H2O | 27.80 | 44.92 | 51.37 | 40.50 | 22.38 | 18.26 | 46.33 | 90.04 | 38.98 | 12.33 | 86.67 | 66.11 | 53.93 | 46.12 |
| | PyramidKV | 30.41 | 47.81 | 51.00 | 41.37 | 22.36 | 17.41 | 61.67 | 90.96 | 37.65 | 13.00 | 86.33 | 67.01 | 50.11 | 47.47 |
| | SnapKV | **32.40** | 49.29 | 54.38 | 41.40 | 23.79 | 18.99 | **63.00** | 91.07 | 38.57 | **14.67** | 86.67 | **67.99** | 53.77 | 48.92 |
| | DapQ | 32.14 | **50.78** | **54.79** | **44.47** | **24.16** | **19.01** | 62.67 | **91.15** | **39.61** | 14.17 | 86.67 | 67.20 | **53.83** | **49.28** |
| | *KV Cache Size = 128* | | | | | | | | | | | | | | |
| | H2O | 26.60 | 41.37 | 48.10 | 39.85 | 20.83 | 17.27 | 41.33 | 90.21 | 38.16 | **11.72** | 86.67 | 65.22 | 52.77 | 44.22 |
| | PyramidKV | 26.22 | 40.48 | 48.41 | 39.46 | 18.99 | 15.10 | 48.33 | 89.31 | 36.92 | 9.67 | 86.67 | 60.82 | 48.78 | 43.78 |
| | SnapKV | 29.41 | 46.43 | 51.20 | 41.66 | 20.37 | 16.64 | 51.33 | **91.07** | 37.37 | 11.00 | 86.67 | 65.36 | 51.65 | 46.17 |
| | DapQ | 29.10 | **47.15** | **53.84** | **43.00** | **21.11** | **17.27** | **54.00** | 90.45 | **38.50** | 11.67 | 86.00 | **66.29** | **52.03** | **46.95** |
| | *KV Cache Size = 64* | | | | | | | | | | | | | | |
| | H2O | 25.55 | 38.94 | 46.66 | 39.27 | **18.55** | **15.23** | 39.00 | 88.13 | 35.98 | 9.67 | **86.67** | 59.48 | **48.95** | 42.47 |
| | PyramidKV | 25.32 | 40.44 | 46.61 | 39.20 | 16.25 | 12.93 | **44.67** | 88.27 | 34.63 | 11.33 | 83.67 | 59.73 | 46.48 | 42.27 |
| | SnapKV | 25.09 | 39.89 | 46.58 | 39.38 | 15.28 | 12.38 | 42.67 | 87.93 | 35.12 | 11.33 | 84.33 | 57.96 | 46.49 | 41.88 |
| | DapQ | **25.78** | **43.17** | **49.84** | **41.28** | 17.16 | 13.95 | 43.00 | **88.97** | 36.00 | **12.67** | 83.00 | **60.31** | 46.90 | **43.23** |

Table 4: Performance comparison of different methods across various LLMs on LongBenchv2. For DapQ, the pseudo-queries are constructed via prefix-suffix concatenation: using the first 8 and last 24 tokens for LLaMA series models, and the first 2 and last 30 tokens for Qwen models. Notably, several compressed methods surpass the FullKV baseline.We attribute this phenomenon to the noise reduction mechanism of cache eviction. By selectively retaining critical tokens, these methods effectively reduce noise and sparsify the context, potentially leading to more focused and efficient model reasoning. This effect is particularly pronounced in long-context benchmarks .

| LLMs | Methods | Difficulty | | Length | | | Overall |
| --- | --- | --- | --- | --- | --- | --- | --- |
| | | Easy | Hard | Short | Medium | Long | |
| | FullKV | 28.65 | 26.37 | 32.22 | 24.19 | 25.00 | 27.24 |
| | KV Cache Size = 128 | | | | | | |
| | H2O | **32.29** | 25.40 | 32.22 | 26.98 | 23.15 | 28.03 |
| | PyramidKV | 30.21 | 24.44 | 31.67 | 26.05 | 19.44 | 26.64 |
| | SnapKV | 30.73 | 25.72 | 32.22 | 26.05 | 23.15 | 27.63 |
| Llama3-8B | DapQ | 30.73 | **27.65** | **33.33** | **27.91** | **23.15** | **28.83** |
| Instruct | KV Cache Size = 64 | | | | | | |
| | H2O | 30.73 | 24.76 | 28.89 | 26.51 | 25.00 | 27.04 |
| | PyramidKV | 27.08 | 23.47 | 24.44 | 27.44 | 20.37 | 24.85 |
| | SnapKV | **31.25** | 22.51 | 23.89 | 26.98 | 26.85 | 25.84 |
| | DapQ | 30.73 | **29.26** | **31.11** | **28.84** | **29.63** | **29.82** |
| | FullKV | 25.00 | 28.62 | 31.67 | 25.58 | 23.15 | 27.24 |
| | KV Cache Size = 128 | | | | | | |
| | H2O | 26.56 | 29.58 | 34.44 | 25.58 | 24.07 | 28.43 |
| | PyramidKV | **29.17** | 28.94 | 32.22 | 27.44 | 26.85 | 29.03 |
| | SnapKV | 27.08 | 29.90 | 33.89 | 26.05 | 25.93 | 28.83 |
| Llama3.1-8B | DapQ | 27.08 | **30.55** | **34.44** | **27.44** | **24.07** | **29.22** |
| Instruct | KV Cache Size = 64 | | | | | | |
| | H2O | 23.44 | 27.01 | 30.56 | 23.72 | 21.30 | 25.65 |
| | PyramidKV | 28.12 | 28.62 | 33.89 | 24.19 | **27.78** | 28.43 |
| | SnapKV | 25.00 | 26.69 | 29.44 | 24.65 | 23.15 | 26.04 |
| | DapQ | **29.17** | **28.94** | **33.89** | 26.98 | 25.00 | **29.03** |
| | FullKV | 28.65 | 27.33 | 30.56 | 27.44 | 24.07 | 27.83 |
| | KV Cache Size = 128 | | | | | | |
| | H2O | 28.65 | **27.65** | 30.56 | 27.91 | **24.07** | 28.03 |
| | PyramidKV | 29.17 | 25.40 | 29.44 | 26.51 | 23.15 | 26.84 |
| | SnapKV | **29.69** | 26.37 | 30.56 | 27.91 | 22.22 | 27.63 |
| Qwen2.5-7B | DapQ | 29.17 | 27.33 | **30.56** | **28.37** | 23.15 | **28.03** |
| Instruct | KV Cache Size = 64 | | | | | | |
| | H2O | 28.65 | 26.37 | 31.11 | 26.51 | 22.22 | 27.24 |
| | PyramidKV | **31.25** | 27.97 | 32.22 | 28.37 | 25.93 | 29.22 |
| | SnapKV | 30.73 | 27.33 | **32.78** | 27.44 | 24.07 | 28.63 |
| | DapQ | 30.73 | **28.30** | 31.67 | **28.37** | **26.85** | **29.22** |
| | FullKV | 31.25 | 28.30 | 33.33 | 25.58 | 30.56 | 29.42 |
| | KV Cache Size = 128 | | | | | | |
| | H2O | 34.38 | 27.97 | 35.56 | 25.58 | **31.48** | 30.42 |
| | PyramidKV | 34.38 | 27.33 | **36.11** | 26.05 | 27.78 | 30.02 |
| | SnapKV | **34.90** | 27.33 | 34.44 | 26.05 | 31.48 | 30.22 |
| Qwen3-8B | DapQ | 33.85 | **28.30** | 33.89 | **27.44** | 30.56 | **30.42** |
| | KV Cache Size = 64 | | | | | | |
| | H2O | 35.42 | 27.65 | 34.44 | 28.84 | 27.78 | 30.62 |
| | PyramidKV | **38.02** | 26.69 | 34.44 | 28.37 | 30.56 | 31.01 |
| | SnapKV | 36.46 | 27.33 | 33.33 | **29.30** | 29.63 | 30.82 |
| | DapQ | 35.94 | **28.62** | **36.67** | 26.51 | **32.41** | **31.41** |

Table 5: Performance comparison of different methods across various kv cache size on Ruler for llama3-8B-Instruct.

| LLM | Methods | Single NIAH | | | Multi-key NIAH | | | MQ-NIAH | MV-NIAH | CWE | FWE | VT | AVG |
|---|---|---|---|---|---|---|---|---|---|---|---|---|---|
| | | S-NIAH-1 | S-NIAH-2 | S-NIAH-3 | MK-NIAH-1 | MK-NIAH-2 | MK-NIAH-3 | | | | | | |
| | FullKV | 100 | 100 | 100 | 99.2 | 91.8 | 95.8 | 99.75 | 97.5 | 97.82 | 82.93 | 98.32 | 96.65 |
| | **KV Cache Size = 2048** | | | | | | | | | | | | |
| | Lacache | 21 | 27.4 | 1.8 | 29.6 | 29 | 3.2 | 12.2 | 6.45 | 86.08 | 86.07 | 11.04 | 28.53 |
| | StreamingLLM | 26.6 | 25.4 | 25.4 | 22.6 | 24.2 | 17.8 | 24.1 | 24.3 | 4.9 | 78.6 | 21.16 | 26.82 |
| | H2O | 100 | 90 | 14 | 78 | 47 | 12.6 | 74.7 | 45 | 90.04 | 79.33 | 97.68 | 66.21 |
| | PyramidKV | 100 | 100 | 40.8 | 99 | 72.2 | 17.4 | 99 | 96.3 | 83.25 | 67.87 | 98.2 | 79.46 |
| | SnapKV | 100 | 98.4 | 61.4 | 99 | 69 | 19.8 | 98.8 | 94.45 | 90.34 | 73.13 | 97.56 | 81.99 |
| | DapQ | 100 | 99 | 96 | 99.2 | 67.2 | 24.4 | 99.05 | 95.45 | 90.68 | 75 | 97.64 | **85.78** |
| | **KV Cache Size = 1024** | | | | | | | | | | | | |
| | Lacache | 0.2 | 4.8 | 2.4 | 4.2 | 4 | 0 | 2.3 | 2.45 | 55.02 | 86.87 | 1.56 | 14.89 |
| | StreamingLLM | 13.4 | 13.8 | 11.4 | 11.4 | 13.2 | 10.8 | 10.95 | 11.3 | 0.38 | 75.07 | 8.16 | 16.35 |
| | H2O | 98.2 | 75.8 | 8 | 62 | 38.8 | 4.8 | 49.05 | 10.6 | 63.4 | 75.07 | 92.2 | 52.54 |
| | PyramidKV | 100 | 98.2 | 6.2 | 98.6 | 47.8 | 2 | 97 | 90.8 | 56.14 | 61.53 | 97.08 | 68.67 |
| | SnapKV | 100 | 96.8 | 15.4 | 98.4 | 44.8 | 3.4 | 96.45 | 87.9 | 65.42 | 64.27 | 96.48 | 69.94 |
| | DapQ | 100 | 98.4 | 85.8 | 99.2 | 41.8 | 6.2 | 97.4 | 92.5 | 65.74 | 69.33 | 96.76 | **77.56** |
| | **KV Cache Size = 512** | | | | | | | | | | | | |
| | Lacache | 0 | 0.2 | 0 | 2.6 | 0.4 | 0 | 0.05 | 1.4 | 16.52 | 78.8 | 0.08 | 9.10 |
| | StreamingLLM | 3.6 | 6.2 | 5.6 | 6.6 | 6.2 | 5.4 | 6.25 | 6.75 | 0.18 | 75.33 | 1.08 | 11.20 |
| | H2O | 88.4 | 63.8 | 2.4 | 45 | 25.4 | 1.4 | 24.4 | 2.85 | 46.56 | 64.6 | 69.64 | 39.50 |
| | PyramidKV | 100 | 95.6 | 0 | 97.4 | 35 | 0.2 | 91.8 | 73.5 | 23.56 | 52.8 | 92.96 | 60.26 |
| | SnapKV | 100 | 95.6 | 1.4 | 96.8 | 30.4 | 0.4 | 91.1 | 71.65 | 30.82 | 53.73 | 94.48 | 60.58 |
| | DapQ | 100 | 97.8 | 59.6 | 98.6 | 29.8 | 1 | 91.95 | 82.9 | 27.16 | 61.27 | 95.2 | **67.75** |
| | **KV Cache Size = 256** | | | | | | | | | | | | |
| | Lacache | 0 | 0 | 0 | 0 | 0 | 0 | 0 | 0 | 3.64 | 58.2 | 0 | 5.62 |
| | StreamingLLM | 1.2 | 1.2 | 1.2 | 2 | 3.4 | 2.4 | 2.3 | 2.45 | 0.14 | 78.6 | 0 | 8.63 |
| | H2O | 67.2 | 56.8 | 2.4 | 25.4 | 15.6 | 0 | 9.05 | 1.25 | 31.1 | 48.2 | 20.64 | 25.24 |
| | PyramidKV | 100 | 94.8 | 0 | 89.6 | 29.4 | 0 | 73.5 | 35.15 | 9.42 | 39.8 | 75.8 | 49.77 |
| | SnapKV | 100 | 95 | 0 | 90.2 | 26 | 0 | 76.5 | 36.3 | 13.66 | 45.2 | 91.56 | 52.22 |
| | DapQ | 100 | 97.6 | 23 | 97.8 | 19.8 | 0 | 79.7 | 55 | 12.8 | 51.87 | 88.04 | **56.87** |
| | **KV Cache Size = 128** | | | | | | | | | | | | |
| | Lacache | 0 | 0 | 0 | 0 | 0 | 0 | 0 | 0 | 0.58 | 8.4 | 0 | 0.82 |
| | StreamingLLM | 0.6 | 1.2 | 1.2 | 2 | 2 | 0 | 2.25 | 2.45 | 0.2 | 44.93 | 0 | 5.17 |
| | H2O | 41.4 | 38.8 | 2.4 | 14.8 | 2.8 | 0 | 2.2 | 0.3 | 18.06 | 13 | 7.88 | 12.88 |
| | PyramidKV | 99.2 | 91 | 0 | 68.2 | 33.2 | 0 | 26.5 | 9.7 | 2.38 | 25.6 | 18.76 | 34.05 |
| | SnapKV | 98.8 | 89 | 0 | 60.2 | 35.4 | 0 | 17.25 | 7.45 | 5.18 | 28.53 | 13.24 | 32.28 |
| | DapQ | 99.6 | 97.6 | 1.4 | 94.4 | 21.4 | 0 | 28.85 | 20.05 | 4.32 | 30.33 | 6.84 | **36.80** |
| | **KV Cache Size = 64** | | | | | | | | | | | | |
| | Lacache | 0 | 0 | 0 | 0 | 0 | 0 | 0 | 0 | 0.18 | 0.67 | 0 | 0.08 |
| | StreamingLLM | 0 | 0 | 0 | 0 | 0 | 0 | 0 | 0 | 0.06 | 27.53 | 0 | 2.51 |
| | H2O | 22 | 21.2 | 0 | 3.8 | 0.2 | 0 | 0.4 | 0.25 | 5.7 | 0.07 | 3.52 | 5.19 |
| | PyramidKV | 48.8 | 47.2 | 0 | 13.4 | 7.2 | 0 | 0.4 | 0.25 | 0.08 | 0 | 0.6 | 10.72 |
| | SnapKV | 58.8 | 65.4 | 0 | 20.4 | 14 | 0 | 0.75 | 0.4 | 0.14 | 0.07 | 2.28 | 14.75 |
| | DapQ | 85.2 | 87.4 | 0 | 26.2 | 19.6 | 0 | 3.65 | 1.35 | 0.78 | 0.33 | 3.6 | **20.74** |

*(LLM column: Llama3-8B-Instruct)*

Table 6: Performance comparison of different methods across various kv cache size on Ruler for Qwen2.5-7B-Instruct.

| LLM | Methods | Single NIAH | | | Multi-key NIAH | | | MQ-NIAH | MV-NIAH | CWE | FWE | VT | AVG |
|---|---|---|---|---|---|---|---|---|---|---|---|---|---|
| | | S-NIAH-1 | S-NIAH-2 | S-NIAH-3 | MK-NIAH-1 | MK-NIAH-2 | MK-NIAH-3 | | | | | | |
| | FullKV | 100 | 99.8 | 99.8 | 99.8 | 98 | 93.2 | 99.8 | 93.9 | 77.38 | 87.67 | 95.36 | 94.97 |
| | **KV Cache Size = 4096** | | | | | | | | | | | | |
| | Lacache | 3 | 1.8 | 2.4 | 4 | 4.8 | 3 | 3.3 | 2.25 | 62.78 | 87.73 | 6.32 | 16.49 |
| | StreamingLLM | 26.4 | 28 | 27 | 24.4 | 19.6 | 11.6 | 26.1 | 27.1 | 36.96 | 91.53 | 22.84 | 31.05 |
| | H2O | 100 | 98.4 | 24 | 96.8 | 19.4 | 9.4 | 85.75 | 68.15 | 67.56 | 92.33 | 93.92 | 68.70 |
| | PyramidKV | 100 | 99.4 | 41.8 | 99.4 | 19.8 | 3.6 | 91.6 | 83.4 | 47.66 | 91.07 | 93.96 | 70.15 |
| | SnapKV | 100 | 99.8 | 86 | 99.8 | 39.6 | 13 | 97.15 | 88.6 | 67.72 | 91.2 | 93.64 | 79.68 |
| | DapQ | 100 | 99.2 | 82 | 99 | 56.8 | 23.6 | 97.25 | 82.15 | 68.04 | 92.53 | 94.52 | **81.37** |
| | **KV Cache Size = 2048** | | | | | | | | | | | | |
| | Lacache | 0.2 | 1.6 | 0 | 2.4 | 0.4 | 0 | 0 | 1.2 | 36.74 | 87.33 | 0.68 | 11.87 |
| | StreamingLLM | 13.8 | 13.4 | 11 | 11.4 | 9 | 6.2 | 10.8 | 11.3 | 5.56 | 94.67 | 9.6 | 17.88 |
| | H2O | 98 | 90.8 | 7.8 | 90.2 | 7.2 | 3.8 | 65.5 | 35 | 55.78 | 93.07 | 89.72 | 57.90 |
| | PyramidKV | 99.4 | 97.2 | 9.2 | 97.8 | 11 | 0.8 | 75.9 | 55 | 28.7 | 94.73 | 96.04 | 60.52 |
| | SnapKV | 100 | 99.2 | 47.6 | 98.8 | 25 | 2.4 | 92.25 | 79.85 | 55.72 | 95.33 | 95.6 | 71.98 |
| | DapQ | 100 | 96.4 | 53 | 97.2 | 43.2 | 7 | 93 | 69.4 | 56.44 | 95.6 | 94.44 | **73.24** |
| | **KV Cache Size = 1024** | | | | | | | | | | | | |
| Qwen2.5-7B-Instruct | Lacache | 0 | 1.6 | 2.4 | 3.2 | 0 | 0 | 1.8 | 2.35 | 13.56 | 84.47 | 0 | 9.94 |
| | StreamingLLM | 4.4 | 6.2 | 5.6 | 6.6 | 4 | 4.2 | 6.2 | 6.7 | 0.26 | 96.93 | 3.16 | 13.11 |
| | H2O | 96.4 | 73.8 | 2.4 | 78.2 | 2.8 | 1.6 | 38.2 | 11.15 | 42.12 | 87.53 | 71.16 | 45.94 |
| | PyramidKV | 99 | 91 | 0.6 | 91.6 | 3.6 | 0 | 48.25 | 25.1 | 12.08 | 95.2 | 93.32 | 50.89 |
| | SnapKV | 99.4 | 98.4 | 14.4 | 97.8 | 13.8 | 0.8 | 78.1 | 58.7 | 37.94 | 96.4 | 92.48 | 62.57 |
| | DapQ | 99.8 | 91.8 | 18.4 | 94.2 | 28.4 | 2.4 | 82.15 | 50.8 | 37.34 | 97 | 93.64 | **63.27** |
| | **KV Cache Size = 512** | | | | | | | | | | | | |
| | Lacache | 0 | 0 | 0 | 0 | 0 | 0 | 0 | 0 | 5 | 71.47 | 0 | 6.95 |
| | StreamingLLM | 1.6 | 1.2 | 1.2 | 2 | 2.2 | 2.6 | 2.3 | 2.4 | 0.26 | 98.87 | 0.36 | 10.45 |
| | H2O | 91 | 53.4 | 2.4 | 52.4 | 0.6 | 0.8 | 14 | 3.75 | 25.64 | 64.87 | 39.92 | 31.71 |
| | PyramidKV | 96.8 | 74.8 | 0 | 62.6 | 1 | 0 | 15.6 | 8.6 | 2.52 | 82.47 | 59.84 | 36.75 |
| | SnapKV | 99 | 89.6 | 1 | 93.8 | 5 | 0.2 | 56.8 | 29.55 | 22.1 | 93.33 | 92.12 | 52.95 |
| | DapQ | 99.6 | 82.8 | 3.4 | 87.4 | 16.6 | 0.4 | 62.4 | 29.75 | 21.9 | 95.2 | 87.52 | **53.36** |
| | **KV Cache Size = 256** | | | | | | | | | | | | |
| | Lacache | 0 | 0 | 0 | 0 | 0 | 0 | 0 | 0 | 1.58 | 21.27 | 0 | 2.08 |
| | StreamingLLM | 0.6 | 1.2 | 1.2 | 3.6 | 0.6 | 0 | 2.3 | 2.4 | 0.22 | 98.07 | 0 | 10.02 |
| | H2O | 63.8 | 22.4 | 2.4 | 13.6 | 0.6 | 0 | 4.25 | 1.5 | 12.46 | 31.13 | 8.76 | 14.63 |
| | PyramidKV | 67.4 | 37.6 | 0 | 16.6 | 0.8 | 0 | 0.75 | 0.6 | 0.36 | 60.67 | 23.96 | 18.98 |
| | SnapKV | 97.6 | 73.4 | 0 | 69.6 | 2 | 0 | 18.95 | 5.8 | 10.52 | 81.6 | 54.84 | 37.66 |
| | DapQ | 98.8 | 63.2 | 0.2 | 71.4 | 10.8 | 0 | 21.4 | 6.35 | 10.7 | 83.73 | 56.96 | **38.5** |
| | **KV Cache Size = 128** | | | | | | | | | | | | |
| | Lacache | 0 | 0 | 0 | 0 | 0 | 0 | 0 | 0 | 0.6 | 2.67 | 0 | 0.30 |
| | StreamingLLM | 0.4 | 1.4 | 0 | 2 | 0.2 | 0 | 2.3 | 2.4 | 0.34 | 96.13 | 0 | 9.56 |
| | H2O | 6 | 3.4 | 0 | 3.6 | 0.2 | 0 | 0.05 | 2.05 | 7.26 | 3.4 | 0.48 | 2.40 |
| | PyramidKV | 4.6 | 6 | 0 | 3.2 | 0 | 0 | 0 | 0 | 0.26 | 21.27 | 2.48 | 3.44 |
| | SnapKV | 57.2 | 30.8 | 0 | 16.8 | 0.6 | 0 | 0.4 | 0.3 | 1.02 | 39.27 | 5.64 | 13.82 |
| | DapQ | 58 | 31.4 | 0 | 39.2 | 3.2 | 0 | 0.6 | 0.9 | 1.46 | 34.73 | 9.96 | **16.31** |
| | **KV Cache Size = 64** | | | | | | | | | | | | |
| | Lacache | 0 | 0 | 0 | 0 | 0 | 0 | 0 | 0 | 0.7 | 0.67 | 0 | 0.12 |
| | StreamingLLM | 0 | 0 | 0 | 0 | 0 | 0 | 0 | 0 | 0.38 | 0 | 0.04 | 0.04 |
| | H2O | 0.2 | 0 | 0 | 0 | 0 | 0 | 0 | 0 | 0.98 | 0 | 0.04 | 0.11 |
| | PyramidKV | 0 | 0 | 0 | 0 | 0 | 0 | 0 | 0 | 0.2 | 0 | 0 | 0.02 |
| | SnapKV | 0 | 0 | 0 | 0.4 | 0 | 0 | 0 | 0 | 0.28 | 0.07 | 0.44 | 0.11 |
| | DapQ | 5.4 | 2.6 | 0 | 3.2 | 0.4 | 0 | 0 | 0 | 0.28 | 2.2 | 1.16 | **1.39** |

Table 7: Performance comparison of different methods across various kv cache size on Ruler for Qwen3-8B.

| LLM | Methods | Single NIAH | | | Multi-key NIAH | | | | | | | | |
| | | S-NIAH-1 | S-NIAH-2 | S-NIAH-3 | MK-NIAH-1 | MK-NIAH-2 | MK-NIAH-3 | MQ-NIAH | MV-NIAH | CWE | FWE | VT | AVG |
|---|---|---|---|---|---|---|---|---|---|---|---|---|---|
| | FullKV | 100 | 100 | 100 | 99.6 | 99.6 | 99.6 | 99.9 | 99.75 | 83.98 | 90.67 | 100 | 97.55 |
| | **KV Cache Size = 4096** | | | | | | | | | | | | |
| | Lacache | 17.6 | 14.6 | 13.2 | 16.8 | 20.2 | 7.4 | 14.75 | 6.2 | 62.46 | 68.47 | 12.48 | 23.11 |
| | StreamingLLM | 26.6 | 28 | 27 | 24.4 | 19.6 | 18.4 | 26.15 | 27.1 | 36.62 | 93 | 22.96 | 31.80 |
| | H2O | 100 | 99.8 | 22 | 99.8 | 84.8 | 32.4 | 99.5 | 89.9 | 46.74 | 93 | 99.92 | 78.90 |
| | PyramidKV | 100 | 100 | 24.8 | 99.6 | 91 | 51.2 | 99.9 | 99.55 | 57.4 | 92.87 | 100 | 83.30 |
| | SnapKV | 100 | 100 | 75.2 | 99.8 | 96 | 58 | 99.9 | 99.75 | 72.7 | 93.27 | 100 | 90.42 |
| | DapQ | 100 | 100 | 96.4 | 99.8 | 93 | 58.4 | 99.9 | 99.85 | 71.82 | 93.6 | 100 | **92.07** |
| | **KV Cache Size = 2048** | | | | | | | | | | | | |
| | Lacache | 2 | 1.6 | 2.4 | 3.4 | 1.2 | 0 | 2.45 | 0.9 | 43.86 | 69.4 | 2.04 | 11.75 |
| | StreamingLLM | 13.8 | 13.4 | 11 | 11.4 | 9.4 | 9 | 10.85 | 11.3 | 10.38 | 95.47 | 9.72 | 18.70 |
| | H2O | 100 | 99.4 | 7.8 | 97 | 64.8 | 10.6 | 94.7 | 48.3 | 28.48 | 94.53 | 99.68 | 67.75 |
| | PyramidKV | 100 | 100 | 1.6 | 100 | 79.4 | 19 | 99.9 | 93.6 | 35.46 | 95.6 | 100 | 74.96 |
| | SnapKV | 100 | 100 | 20 | 100 | 91.6 | 30.4 | 99.85 | 97.75 | 49.59 | 96 | 100 | 80.47 |
| | DapQ | 100 | 100 | 55 | 100 | 88.05 | 31 | 99.95 | 95.15 | 45.76 | 96.07 | 100 | **82.82** |
| | **KV Cache Size = 1024** | | | | | | | | | | | | |
| | Lacache | 0.2 | 1.6 | 2.4 | 3.2 | 0.4 | 0 | 2.45 | 1.05 | 15.86 | 62.07 | 0.32 | 8.14 |
| | StreamingLLM | 4.6 | 6.2 | 5.6 | 6.6 | 3.8 | 5 | 6.25 | 6.7 | 0.8 | 96.47 | 3.24 | 13.21 |
| | H2O | 97.8 | 92 | 2.4 | 81.6 | 38.2 | 2.4 | 72 | 15.85 | 23.98 | 96.2 | 93.92 | 56.03 |
| | PyramidKV | 100 | 99.8 | 0 | 98.6 | 61.8 | 2.4 | 97.75 | 70.2 | 17.84 | 92.47 | 99.12 | 67.27 |
| | SnapKV | 100 | 99.6 | 1 | 99.6 | 79.2 | 13 | 99.25 | 83.05 | 26.52 | 96.27 | 98.84 | 72.39 |
| | DapQ | 100 | 99.8 | 14.2 | 99.6 | 75.4 | 14.8 | 99.75 | 78.25 | 23.78 | 97.73 | 99.12 | **72.95** |
| | **KV Cache Size = 512** | | | | | | | | | | | | |
| | Lacache | 0 | 1.6 | 0 | 3 | 0 | 0 | 0 | 0.8 | 7.06 | 26.07 | 0.08 | 3.51 |
| | StreamingLLM | 1.8 | 4 | 1.2 | 3.8 | 2.4 | 3 | 3.75 | 4.4 | 0.76 | 97.6 | 0.56 | 11.21 |
| | H2O | 90.4 | 66.2 | 2.4 | 57.2 | 15 | 0.8 | 30.6 | 5.1 | 14.8 | 89.2 | 65.56 | 39.75 |
| | PyramidKV | 99 | 97.2 | 0 | 84.4 | 40.8 | 0.2 | 70.7 | 30 | 6.2 | 70.27 | 70.92 | 51.79 |
| | SnapKV | 99.6 | 99.2 | 0 | 97.2 | 60.6 | 2 | 94.35 | 47.25 | 14.78 | 93.6 | 98.64 | 64.29 |
| | DapQ | 100 | 99.6 | 5.8 | 95.4 | 68.6 | 5.2 | 94.85 | 42.45 | 12.38 | 96.27 | 96.08 | **65.15** |
| | **KV Cache Size = 256** | | | | | | | | | | | | |
| | Lacache | 0 | 0 | 0 | 0 | 0 | 0 | 0 | 0 | 1.7 | 7.93 | 0 | 0.88 |
| | StreamingLLM | 0.8 | 1.2 | 1.2 | 2 | 2.4 | 0 | 2.3 | 2.4 | 0.72 | 95 | 0 | 9.82 |
| | H2O | 68 | 21.4 | 2.4 | 19 | 2.6 | 0 | 6.15 | 1.4 | 11.2 | 61.27 | 19.2 | 19.33 |
| | PyramidKV | 94.8 | 60.4 | 0 | 41.6 | 14.4 | 0 | 5.8 | 5.6 | 1.22 | 34.07 | 24 | 25.63 |
| | SnapKV | 97.4 | 87.8 | 0 | 81.6 | 35.2 | 0 | 47.2 | 13 | 9.7 | 84.47 | 72.32 | 48.06 |
| | DapQ | 100 | 88.8 | 0 | 75.4 | 59.4 | 0.2 | 47.55 | 10.2 | 6.1 | 88 | 58.04 | **48.52** |
| | **KV Cache Size = 128** | | | | | | | | | | | | |
| | Lacache | 0 | 0 | 0 | 0 | 0 | 0 | 0 | 0 | 0.46 | 0.2 | 0 | 0.06 |
| | StreamingLLM | 0.6 | 1.2 | 1.2 | 2 | 0.2 | 0 | 2.3 | 2.4 | 0.64 | 96 | 0 | 9.69 |
| | H2O | 19.8 | 2.4 | 1.6 | 3.6 | 0.2 | 0 | 0.05 | 0.25 | 8.4 | 22.53 | 3.44 | 5.66 |
| | PyramidKV | 11.4 | 2.8 | 0 | 0 | 1.2 | 0 | 0 | 0 | 0.84 | 4.6 | 3.68 | 2.23 |
| | SnapKV | 68.8 | 31.2 | 0 | 2.6 | 3.2 | 0 | 0.15 | 0.45 | 2.06 | 40.33 | 5.24 | 14.00 |
| | DapQ | 97.8 | 34 | 0 | 7 | 20.8 | 0 | 0.2 | 0.1 | 1.1 | 40.53 | 5.47 | **18.82** |
| | **KV Cache Size = 64** | | | | | | | | | | | | |
| | Lacache | 0 | 0 | 0 | 0 | 0 | 0 | 0 | 0 | 0.72 | 0 | 0 | 0.07 |
| | StreamingLLM | 0 | 0 | 0 | 0 | 0 | 0 | 0 | 0 | 0.4 | 32.33 | 0 | **2.98** |
| | H2O | 0 | 0 | 0 | 0 | 0 | 0 | 0 | 0 | 3.42 | 0 | 1.24 | 0.42 |
| | PyramidKV | 0 | 0 | 0 | 0.2 | 0 | 0 | 0 | 0 | 0.78 | 0.4 | 0.56 | 0.18 |
| | SnapKV | 0 | 0 | 0 | 0 | 0 | 0 | 0 | 0 | 0.8 | 0.07 | 0.88 | 0.16 |
| | DapQ | 0.8 | 0.2 | 0 | 0 | 1.2 | 0 | 0 | 0 | 1 | 2.4 | 1.68 | 0.66 |

*(LLM column: Qwen3-8B)*

Table 8: Performance comparison of different methods across various LLMs on sub-task categories of the HELMET benchmark.

| | Methods | ICL (exact_match) | | | | | LONGQA | | RAG (substring_exact_match) | | | | Avg. |
|---|---|---|---|---|---|---|---|---|---|---|---|---|---|
| | | | | | | | f1 | rougeL_f1 | | | | | |
| | | icl_banking77 | icl_clinic150 | icl_nlu | icl_trec_coarse | icl_trec_fine | narrativeqa | infbench_qa_eng | kilt_hotpotqa | kilt_nq | kilt_popqa_3 | kilt_triviaqa | |
| **Llama3-8B-Instruct** | FullKV | 38.60 | 73.60 | 76.20 | 39.40 | 25.00 | 12.00 | 16.56 | 52.00 | 42.17 | 47.67 | 80.00 | 45.75 |
| | **KV Cache Size = 1024** | | | | | | | | | | | | |
| | H2O | **32.00** | 64.00 | 70.60 | 36.00 | **21.20** | 11.30 | 15.28 | 47.67 | 44.17 | 47.67 | 81.50 | 42.85 |
| | PyramidKV | 24.80 | 55.40 | 68.60 | 29.60 | 16.00 | 11.61 | 16.00 | 52.33 | 44.17 | 48.17 | 81.33 | 40.73 |
| | SnapKV | 23.00 | 54.40 | 70.60 | 29.00 | 18.20 | 11.04 | 17.07 | **52.67** | 43.83 | 48.33 | 81.67 | 40.89 |
| | DapQ | 26.80 | **64.40** | **72.60** | **39.80** | 15.40 | **11.81** | 16.67 | 50.00 | **45.00** | **48.33** | **81.83** | **42.97** |
| | **KV Cache Size = 512** | | | | | | | | | | | | |
| | H2O | 18.00 | 51.80 | 63.00 | 28.80 | 18.00 | 10.65 | 14.45 | 48.33 | 44.17 | 48.00 | 82.00 | 38.84 |
| | PyramidKV | 17.40 | 39.80 | 56.00 | 22.20 | 12.20 | **11.69** | 15.87 | 50.67 | 44.17 | 48.00 | 82.17 | 36.38 |
| | SnapKV | 17.60 | 41.40 | 63.60 | 22.20 | 14.20 | 11.34 | 16.67 | **50.67** | 44.50 | 47.17 | 82.83 | 37.47 |
| | DapQ | **21.60** | **52.80** | **64.00** | **39.20** | **15.40** | 11.18 | **16.89** | 49.00 | **46.00** | 48.17 | 82.17 | **40.58** |
| | **KV Cache Size = 256** | | | | | | | | | | | | |
| | H2O | 11.60 | 35.40 | **54.20** | 23.40 | 11.20 | 11.21 | 12.82 | 46.67 | 42.33 | 47.00 | 80.67 | 34.23 |
| | PyramidKV | 14.00 | 27.40 | 36.20 | 17.00 | 10.00 | 11.36 | 14.94 | **52.00** | 42.83 | 48.00 | 81.17 | 32.26 |
| | SnapKV | 16.80 | 27.80 | 47.60 | 17.80 | 9.00 | 11.52 | 14.76 | 50.67 | 42.67 | 46.83 | 81.67 | 33.37 |
| | DapQ | **17.00** | **37.40** | 47.00 | **38.40** | 13.80 | **11.66** | 15.70 | 48.67 | **44.00** | 48.17 | 81.83 | **36.69** |
| | **KV Cache Size = 128** | | | | | | | | | | | | |
| | H2O | 6.80 | 20.00 | **32.80** | 18.00 | 6.20 | 10.76 | 12.52 | 45.33 | 41.00 | 46.00 | 81.33 | 29.16 |
| | PyramidKV | 11.20 | 21.80 | 19.60 | 14.80 | 8.20 | 10.70 | 14.03 | 48.00 | 39.33 | 46.50 | 83.50 | 28.88 |
| | SnapKV | 13.80 | 19.60 | 21.40 | 14.60 | 8.80 | 10.92 | 13.66 | 46.00 | 38.83 | 47.07 | 84.33 | 29.00 |
| | DapQ | **16.40** | **23.40** | 25.60 | **31.00** | **13.20** | **10.99** | **15.14** | **49.33** | **41.50** | **47.33** | **84.67** | **32.60** |
| **Qwen2.5-7B-Instruct** | FullKV | 74.00 | 71.00 | 53.80 | 75.60 | 31.80 | 20.53 | 30.33 | 56.00 | 49.83 | 57.67 | 86.67 | 55.20 |
| | **KV Cache Size = 2048** | | | | | | | | | | | | |
| | H2O | 63.20 | 54.00 | 51.00 | **79.40** | 31.40 | **20.37** | 28.74 | 52.00 | **49.33** | 58.17 | 87.00 | 52.24 |
| | PyramidKV | 68.40 | 61.40 | 53.20 | 77.40 | 33.20 | 19.46 | 28.52 | 53.67 | 48.00 | **60.83** | 85.33 | 53.58 |
| | SnapKV | 68.40 | 62.00 | 51.40 | 78.00 | 33.20 | 20.35 | 29.41 | 55.67 | 47.83 | 58.33 | 86.67 | 53.75 |
| | DapQ | **71.00** | **67.80** | **55.00** | 76.00 | **33.60** | 19.05 | 29.46 | **56.33** | 48.83 | 58.50 | **87.00** | **54.78** |
| | **KV Cache Size = 1024** | | | | | | | | | | | | |
| | H2O | 44.60 | 33.20 | 28.00 | **79.40** | 25.40 | 19.52 | 27.24 | 50.67 | **48.50** | 57.00 | 85.50 | 45.37 |
| | PyramidKV | 58.40 | 40.60 | 39.00 | 78.20 | 29.20 | 20.30 | 28.11 | 50.67 | 44.50 | **61.00** | 84.50 | 48.59 |
| | SnapKV | 56.80 | 42.10 | 38.00 | 75.00 | 30.80 | **20.53** | 28.05 | 56.00 | 48.00 | 58.33 | 85.37 | 49.00 |
| | DapQ | **66.00** | **56.40** | **49.80** | 76.20 | **31.20** | 20.20 | **28.76** | 54.33 | 46.67 | 57.50 | **85.67** | **52.07** |
| | **KV Cache Size = 512** | | | | | | | | | | | | |
| | H2O | 28.60 | 19.40 | 17.00 | **76.20** | 17.80 | 18.51 | 26.88 | 51.27 | 45.33 | 57.67 | 85.33 | 40.36 |
| | PyramidKV | 45.00 | 20.00 | 22.80 | 70.60 | 25.80 | **21.91** | 26.60 | 48.67 | 43.50 | **59.67** | 82.83 | 42.49 |
| | SnapKV | 44.60 | 23.60 | 22.20 | 71.80 | 26.80 | 20.17 | 27.85 | 53.00 | **47.17** | 58.17 | 85.83 | 43.74 |
| | DapQ | **52.00** | **43.20** | **40.80** | 72.20 | **28.40** | 20.54 | **29.50** | **54.00** | 45.33 | 57.33 | 85.83 | **48.10** |
| | **KV Cache Size = 256** | | | | | | | | | | | | |
| | H2O | 21.20 | 12.00 | 9.20 | **74.40** | 14.40 | 18.26 | 26.12 | 48.67 | 43.00 | 58.67 | 83.33 | 37.20 |
| | PyramidKV | 34.80 | 14.00 | 13.20 | 50.80 | 15.00 | 17.36 | 25.93 | 45.00 | 40.00 | 59.17 | 75.17 | 35.49 |
| | SnapKV | 38.40 | 15.20 | 18.80 | 60.00 | 19.80 | **19.90** | 26.55 | 51.00 | **45.33** | **59.83** | 82.17 | 39.73 |
| | DapQ | **41.40** | **25.40** | **27.60** | 67.40 | **21.00** | 17.64 | **26.55** | **51.33** | 47.00 | 59.00 | **85.17** | **42.68** |

Table 9: Performance comparison of different methods across various LLMs on Needle-in-a-Haystack.

| LLM | KV Cache Size | Method | Acc |
|---|---|---|---|
| | | **FullKV** | **100.00** |
| | 256 | H2O | 66.81 |
| | | PyramidKV | 93.94 |
| | | SnapKV | 90.97 |
| | | **DapQ** | **99.46** |
| Llama3-8B-Instruct | 128 | H2O | 50.92 |
| | | PyramidKV | 79.67 |
| | | SnapKV | 74.67 |
| | | **DapQ** | **95.75** |
| | 64 | H2O | 42.37 |
| | | PyramidKV | 55.45 |
| | | SnapKV | 61.70 |
| | | **DapQ** | **68.34** |
| | | **FullKV** | **98.02** |
| | 256 | H2O | 61.18 |
| | | PyramidKV | 78.30 |
| | | SnapKV | 74.84 |
| | | **DapQ** | **84.70** |
| Llama3.1-8B-Instruct | 128 | H2O | 47.61 |
| | | PyramidKV | 65.23 |
| | | SnapKV | 61.45 |
| | | **DapQ** | **70.34** |
| | 64 | H2O | 40.36 |
| | | PyramidKV | 52.25 |
| | | SnapKV | 56.50 |
| | | **DapQ** | **62.20** |
| | | **FullKV** | **94.23** |
| | 256 | H2O | 75.64 |
| | | PyramidKV | 83.80 |
| | | SnapKV | 84.30 |
| | | **DapQ** | **85.11** |
| Qwen2.5-7B-Instruct | 128 | H2O | 70.45 |
| | | PyramidKV | 74.80 |
| | | SnapKV | 73.64 |
| | | **DapQ** | **76.25** |
| | 64 | H2O | 63.70 |
| | | PyramidKV | 56.11 |
| | | SnapKV | 72.84 |
| | | **DapQ** | **75.75** |
| | | **FullKV** | **96.52** |
| | 256 | H2O | 74.55 |
| | | PyramidKV | 88.50 |
| | | SnapKV | 90.41 |
| | | **DapQ** | **91.73** |
| Qwen3-8B | 128 | H2O | 67.50 |
| | | PyramidKV | 72.36 |
| | | SnapKV | 75.39 |
| | | **DapQ** | **77.89** |
| | 64 | H2O | **62.70** |
| | | PyramidKV | 61.32 |
| | | SnapKV | 59.73 |
| | | **DapQ** | 61.98 |

