# OpenReview forum: "Where Matters More Than What: Decoding-aligned KV Cache Compression via Position-aware Pseudo-queries"
_ICLR.cc/2026/Conference — ICLR 2026 Conference Withdrawn Submission_

### Official Review · Reviewer_dTuz · 2025-10-26

**Soundness:** 2
**Presentation:** 2
**Contribution:** 2
**Rating:** 4
**Confidence:** 4

**Summary:**

This work proposes a decoding-aligned KV cache compression method that estimates important tokens for future decoding by constructing position-aware pseudo-queries. The key finding is that positional encoding contributes more to query similarity than semantic content, shows that the location of a token matters more than its meaning when determining attention patterns. Based on this insight, the authors design DapQ, which appends synthetic pseudo-tokens after the input prompt to simulate future decoding steps. These pseudo-queries are then used to compute future attention to existing prompt tokens, allowing the model to estimate token importance and selectively retain only the most critical tokens in the KV cache eviction framework.

**Strengths:**

* **Novel Insight on Query Similarity.** This work presents a compelling new finding that positional encoding, rather than semantic content, primarily determines query representations. This insight reshapes our understanding of how attention alignment and KV cache compression should be approached.

**Weaknesses:**

* **Fixed Prefill-Only Compression.** DapQ performs token eviction only during the prefill stage. Its efficiency and effectiveness in long-context generation scenarios, such as chain-of-thought reasoning or multi-step tasks, remain unexplored.
* **Lack of Efficiency Evaluation.** The proposed method introduces additional pseudo-tokens during prefill, effectively enlarging the input length and increasing the Time to First Token (TTFT). As attention computation scales with $O(N^2)$ in the prefill phase, the overhead may become significant for long inputs. Since KV cache compression is mainly designed to improve efficiency, runtime and latency evaluations should be included to demonstrate the actual benefits versus the added cost.
* **Unclear Generalization Across Positional Embeddings.** The proposed method is designed around RoPE-based models. However, it is unclear how well it generalizes to models using NoPE (e.g., Jamba1.5 [1]) or NTK-scaled RoPE (as LLaMA3-8B and Qwen3-8B provided). The behavior and compatibility of DapQ under different positional encoding schemes require further clarification.
* **Unexpected Benchmark Results.** In the LongBench `Code` tasks with `LLaMA3-8B-Instruct`, several compression methods unexpectedly outperform the Full KV cache baseline. This phenomenon warrants additional analysis to explain why cache eviction improves over the uncompressed baseline. Moreover, since Qwen3-8B offers an optional reasoning mode, the paper should clarify whether this mode was enabled during evaluation, as it may influence performance outcomes.

[1] Team, Jamba, et al. "Jamba-1.5: Hybrid transformer-mamba models at scale." *arXiv preprint arXiv:2408.12570* (2024).

**Questions:**

* As noted in the weaknesses, how does DapQ perform on reasoning-oriented models? Could the authors provide comparisons between Qwen3-8B with and without reasoning mode to illustrate potential differences in long-context reasoning performance?

---

> ### Author Response · Authors · 2025-11-24
>
> ***W1:*** Fixed Prefill-Only Compression. DapQ performs token eviction only during the prefill stage. Its efficiency and effectiveness in long-context generation scenarios, such as chain-of-thought reasoning or multi-step tasks, remain unexplored.
>
>   ***A1:*** We deeply appreciate the reviewer raising this insightful and crucial point. It is of significant importance for clarifying DapQ's scope of application and further refining the method.
>
> You notes that DapQ currently executes compression (token eviction) only once during the prefill stage, leaving its efficiency and effectiveness in long decoding tasks unexplored. We fully acknowledge this concern and wish to provide further clarification here. **The core idea of DapQ is entirely applicable to the decoding stage and is not inherently limited to being "prefill-only."**
>
> Our current validation focused primarily on the prefill stage to address the most significant KV Cache bottleneck encountered in long-context scenarios (where KV length is often already very large upon completing prefill). This was the central design scenario for DapQ. However, we completely agree with the reviewer's view: investigating DapQ compression during **long generation tasks** (such as CoT reasoning, long story generation, or multi-step tasks) is highly valuable.
>
> We now explicitly add that DapQ can be extended into **dynamic compression** during the autoregressive decoding process. For instance:During autoregressive decoding, whenever the accumulated KV Cache length reaches a preset capacity threshold, we trigger a compression operation. At this point, we construct a pseudo-query window of size $N$. We set the position IDs of these pseudo-queries to correspond to *future* positions starting immediately from the current decoding step. By utilizing these pseudo-queries aligned with future positions, we calculate importance scores for all accumulated tokens in the past KV Cache. Based on these scores, we evict tokens of lower importance, compressing the KV Cache back to a smaller size to make room for new incoming tokens. Decoding then continues. This operation repeats cyclically.
>
> We have explicitly listed "implementing and evaluating dynamic DapQ compression strategies in long decoding tasks (e.g., story generation, code completion, etc.)" as important **future work**. Thank you again for your valuable feedback, which has provided a clear direction for improving and expanding our work.
>
>
> ***
>
>
>
>   ***W2:*** Lack of Efficiency Evaluation. The proposed method introduces additional pseudo-tokens during prefill, effectively enlarging the input length and increasing the TTFT. As attention computation scales with $O(n^2)$  in the prefill phase, the overhead may become significant for long inputs. Since KV cache compression is mainly designed to improve efficiency, runtime and latency evaluations should be included to demonstrate the actual benefits versus the added cost.
>
>   ***A2:*** Thanks for your valuable comments regarding efficiency and latency evaluations.
>
> (1) ***We have supplemented our experimental results with a complete evaluation of Time to First Token (TTFT(s)) on LLaMA-3.1-8B-Instruct, across varying sequence lengths (from 8K to 128K)***, as shown in Table below. It can be observed from the table that the pseudo-tokens introduced by DapQ have a minimal impact on the attention forward computation, resulting in negligible latency overhead.
>
>
> | Method | 8K | 16k | 32k | 64k | 128k |
> | :--- | ---: | ---: | ---: | ---: | ---: |
> | Fullkv | 1.1087 | 2.5576 | 6.5602 | 18.9097 | 61.2364 |
> | Snapkv | 1.1262 | 2.5891 | 6.6218 | 19.0281 | 61.4849 |
> | DapQ | 1.1405 | 2.5925 | 6.6411 | 19.0432 | 61.4922 |
>
>
>
> (2) We acknowledge that pseudo-tokens increase the input length, thereby affecting the one-time prefill stage, which has a complexity of $O(n^2)$. This concern is valid. We further analyzed the theoretical overhead based on the number of pseudo-tokens in DapQ (which is fixed at 32). **As the sequence length $n$ increases, the relative additional overhead decays rapidly (scaling approximately with $1/n$) and quickly approaches zero.**
>
> $$
> \frac{(n + 32)^2}{n^2} = \frac{n^2 + 64n + 1024}{n^2} = 1 + \frac{64}{n} + \frac{1024}{n^2}
> $$
>
> In summary, through both theoretical analysis and empirical measurements, we have demonstrated that the additional overhead caused by the pseudo-tokens in DapQ rapidly diminishes to a negligible level under long sequence conditions. Therefore, even accounting for the prefill overhead, DapQ’s overall efficiency remains significantly superior, and it does not lead to latency degradation in practical systems.
>
>
> （3）And we have supplemented detailed comparative experiments on memory usage and throughput using Llama3.1-8B-Instruct (Input 8k, Output 150 tokens, Budget=256). The experiments demonstrate that while significantly improving long-context understanding capabilities, DapQ maintains high inference efficiency and low memory usage.

---

> > ### Author Response · Authors · 2025-11-24
> >
> > **(2) Regarding Generalization to NTK-scaled RoPE Models**
> >
> > We would like to offer clarification and additional context regarding the reviewer's mention of NTK-scaled RoPE models. In fact, the **LLaMA-3-8B-instruct** and **Qwen-3-8B** models, which we evaluated extensively in our paper, are typical representative models that widely adopt RoPE and its variants (including **NTK-aware scaling** to support long contexts). Our experimental results (Paper Tables 2, 5, 6, and 7) already strongly demonstrate the effectiveness of DapQ on these models.
> >
> >
> > ***
> >
> >
> >
> >   ***W4:*** Unexpected Benchmark Results. In the LongBench Code tasks with LLaMA3-8B-Instruct, several compression methods unexpectedly outperform the Full KV cache baseline. This phenomenon warrants additional analysis to explain why cache eviction improves over the uncompressed baseline. Moreover, since Qwen3-8B offers an optional reasoning mode, the paper should clarify whether this mode was enabled during evaluation, as it may influence performance outcomes.
> >
> >   ***A4:***
> >
> > (1) We thank the reviewer for this keen observation regarding this interesting and counterintuitive phenomenon. Concerning the scenario in tasks like LCC (Long Code Completion) where "compression methods outperform the Full KV baseline," we conducted in-depth code reviews and theoretical analysis, confirming that the experimental results are genuine and reproducible. We believe this is not an error, but rather reflects a **"Denoising Effect"** of KV Cache compression in specific scenarios:
> >
> > *   **① Attention Dilution & Noise Interference:** In extremely long contexts (such as the LCC dataset), the input sequence often contains a large amount of redundant code, comments, or irrelevant function definitions unrelated to the current generation task. With the Full KV Cache, the model must distribute attention probabilities across all these tokens. This leads to "attention dilution" and makes the model susceptible to interference from irrelevant "noise tokens" (i.e., the distraction phenomenon).
> >
> > *   **② Denoising & Focusing via Eviction:** The core mechanism of effective compression baselines is filtering based on token importance. KV cache compression methods can precisely lock onto dependencies most relevant to the current prediction (e.g., key variable definitions, function interfaces). By evicting a large number of low-scoring KV pairs, they filter out noise interference from the context. This allows the Softmax probability mass to be more concentrated on truly critical tokens, thereby assisting the model in making more accurate predictions.
> >
> > *   **③ Impact of Model Characteristics:** Although LLaMA3-8B-Instruct possesses long-window capabilities, without targeted long-context finetuning, it is prone to the "Lost-in-the-middle" problem when facing code contexts of 8K+ or longer with Full KV. In contrast, the compressed KV Cache provides a shorter and more refined context, which paradoxically reduces the reasoning difficulty for the model.
> >
> > We once again thank the reviewer for this suggestion. We will include a discussion regarding this phenomenon in the paper's appendix.
> >
> > (2) We thank the reviewer for pointing out this important detail regarding the experimental setup. In all evaluation processes, the optional Reasoning Mode of Qwen3-8B was **disabled**.
> >
> > To avoid any ambiguity, we will update the "Implementation Details" section in the paper to explicitly state that the reasoning mode was disabled.
> >
> > We chose to disable the reasoning mode because enabling it (which typically triggers extensive Chain-of-Thought generation) introduces significant variables into decoding length and generated content. This makes it difficult to isolate the direct impact of our compression method and baseline methods on retrieval accuracy and context retention capabilities.
> >
> >
> > ***
> >
> >
> >   ***Q1:*** As noted in the weaknesses, how does DapQ perform on reasoning-oriented models? Could the authors provide comparisons between Qwen3-8B with and without reasoning mode to illustrate potential differences in long-context reasoning performance?
> >
> >   ***A5:*** Thanks for raising this insightful question.
> >
> > **(1) Experimental Setup for Reasoning Capability Analysis**
> >
> > To investigate the impact of reasoning capability on long-context compression performance, we compared the performance of Qwen3-8B under two different settings:
> >
> > **① Reasoning OFF:**
> > *   **Max Input Length:** 16k.
> > *   **Output Settings:** We strictly followed the official configuration files for each dataset in LongBench. These usually set a shorter `max_new_tokens` tailored to specific tasks, forcing the model to generate direct answers rather than employing a Chain-of-Thought (CoT).

---

> > > ### Author Response · Authors · 2025-11-24
> > >
> > > **② Reasoning ON:**
> > > *   **Max Input Length:** 16k.
> > > *   **Output Settings:** Considering that enabling reasoning mode prompts the model to perform multi-step deductions via Chain-of-Thought (CoT), the output length increases significantly. To prevent answer truncation due to output length limitations—especially given potential uncertainties introduced by KV Cache compression during the prefill stage, as well as the possibility of getting stuck in repetitive loops within `<think>` and `</think>` tags—we uniformly adjusted the maximum output length for all datasets to **1536 tokens** (allocating approximately 1024 tokens for the reasoning process and 512 tokens for the final answer). This setting aims to ensure the model has sufficient generation space to demonstrate its reasoning process, thereby validating DapQ's effectiveness in preserving critical reasoning cues.
> > >
> > > |Settings|Method|Qasper|MF-en|HotpotQA|2WikiMQA|GovReport|Multi_News|TREC|TriviaQA|SAMSum|LCC|RB-P|PRe|PCount|AVG|
> > > |:---|:---|:---|:---|:---|:---|:---|:---|:---|:---|:---|:---|:---|:---|:---|:---|
> > > |Reasoning OFF|Fullkv|36.87|53.37|57.67|44.73|33.39|23.69|71.67|91.79|42.07|70.64|59.26|86.67|11.98|52.60|
> > > ||Snapkv|32.40|49.29|54.38|41.40|23.79|18.99|63.00|91.07|38.57|67.99|53.77|86.67|14.67|48.92|
> > > ||DapQ|32.14|50.78|54.79|44.47|24.16|19.01|62.67|91.15|39.61|67.20|53.83|86.67|14.17|49.28|
> > > |Reasoning ON|Fullkv|40.00|48.28|69.97|74.24|29.40|20.31|48.02|88.51|33.15|13.67|13.83|99.56|31.15|46.93|
> > > ||Snapkv|31.51|45.10|59.56|47.54|21.31|16.97|43.83|87.50|31.65|10.18|7.24|89.67|16.46|39.12|
> > > ||DapQ|32.83|46.71|60.93|53.35|23.34|17.51|44.44|87.42|32.08|11.92|8.83|89.48|18.31|40.55|
> > >
> > >
> > > **(2) Potential Differences in Long-Context Reasoning Performance**
> > >
> > > Our findings indicate that the reasoning mode elevates the performance ceiling for complex tasks but demands higher accuracy from the KV Cache. Under reasoning mode, DapQ proves to be more robust compared to SnapKV.
> > >
> > > **Note regarding the performance drop in LCC and RepoBench-P under reasoning mode:**
> > > Upon enabling reasoning mode, we observed an abnormally sharp decline in scores for code completion tasks (LCC, RepoBench-P); for example, LCC dropped from ~70 to ~10. After a thorough investigation, we determined that this decline does not represent a loss of model capability. Instead, it is caused by an incompatibility between the "output format of the reasoning mode" and the "strict matching mechanism of the evaluation metric." The metric used in the experiments is `code_sim_score`. With Reasoning ON, the model tends to provide explanations first or wrap code blocks in Markdown formatting (e.g., "To complete the code, we need to..."). Since the evaluation script was not adapted for CoT formats (e.g., extracting Markdown code blocks), the calculated similarity scores approached zero. In reality, manual inspection of cases revealed that the model still correctly identified the code logic under reasoning mode, often providing even more detailed explanations; however, its output form did not meet the preset input requirements of that specific metric.

---

> > > > ### Comment · Reviewer_dTuz · 2025-11-27
> > > > **Reply to Author's Response 3&4.**
> > > >
> > > > * To A4. The response should an interesting finding that kv cache eviction can achieve a denoising effect.
> > > > * To A5. Thanks the author's experiment on hybrid reasoning and analysis on performance drops.

---

> > ### Comment · Reviewer_dTuz · 2025-11-27
> > **Reply to Author's Response 1.**
> >
> > * To A1. Thanks for your notice on decoding steps. Hope your future implementation achieve better results.
> > * To A2. The absolute additional overhead will increase O(n) in theory. And for experiment side, may I know the inference backend and your devices on FullKV? As my test on Llama3.1-8B, with even single 3090 and vllm with flash-attention backend, the TTFT latency of FullKV is like 10x-30x smaller than your experiments:
> > | 8K      | 16K      | 32K      |
> > | ------- | -------- | -------- |
> > | 89.94ms | 137.74ms | 226.30ms |

---

> ### Author Response · Authors · 2025-11-24
>
> ### Comparison of Memory (GB) with Different Batch Sizes
>
> |Method|Batch_size=1|Batch_size=10|Batch_size=20|Batch_size=30|Batch_size=40|Batch_size=50|
> |:---|:---:|:---:|:---:|:---:|:---:|:---:|
> |**Fullkv**|16.87|33.74|52.49|71.24|OOM|OOM|
> |**SLM**|16.01|25.17|35.36|45.55|55.73|65.92|
> |**Lacache**|16.87|33.74|52.49|71.24|OOM|OOM|
> |**Pyramidkv**|16.01|25.23|35.47|45.72|55.96|66.20|
> |**Snapkv**|16.01|25.17|35.36|45.55|55.73|65.92|
> |**DapQ**|16.01|25.21|35.43|45.65|55.87|66.02|
>
>
> ### Comparison of throughput (tokens/s) with different batch sizes
>
> |Method|Batch_size=1|Batch_size=10|Batch_size=20|Batch_size=30|Batch_size=40|Batch_size=50|
> |:---|:---:|:---:|:---:|:---:|:---:|:---:|
> |**Fullkv**|11.59|26.43|25.60|26.59|OOM|OOM|
> |**SLM**|10.81|34.49|38.21|39.25|39.98|40.46|
> |**Lacache**|11.44|35.77|40.64|42.49|OOM|OOM|
> |**Pyramidkv**|10.54|34.12|38.00|39.03|39.86|40.11|
> |**Snapkv**|10.77|34.22|38.09|39.10|39.77|40.23|
> |**DapQ**|10.68|34.16|37.99|38.97|39.73|40.12|
>
>
> ***
>
>
>
>   ***W3:*** Unclear Generalization Across Positional Embeddings. The proposed method is designed around RoPE-based models. However, it is unclear how well it generalizes to models using NoPE (e.g., Jamba1.5 [1]) or NTK-scaled RoPE (as LLaMA3-8B and Qwen3-8B provided). The behavior and compatibility of DapQ under different positional encoding schemes require further clarification.
>
>   ***A3:*** We thank the reviewer for the insightful question regarding the generalization of our method across different positional encoding schemes, specifically NoPE and NTK-scaled RoPE.
>
> **(1) Regarding NoPE / Hybrid Architecture Models (taking Jamba-1.5 as an example)**
>
> We have deeply examined the architectural characteristics of Jamba-1.5 and conducted additional experiments on Jamba-1.5-Mini, similar to the setup in Table 1 of our paper.
>
> **① Jamba-1.5 Architecture and KV Cache:** Jamba-1.5 adopts a novel hybrid architecture (**Hybrid Transformer-Mamba**), characterized by interleaving Transformer layers with Mamba (SSM) layers. Notably, the ratio of **Attention layers to Mamba layers is only 1:7**. This means the Jamba model requires storing KV Cache in only a very small number of Attention layers, while Mamba layers, operating as State Space Models (SSMs), do not rely on the traditional KV Cache mechanism. Consequently, Jamba-1.5 itself has **tremendously compressed memory usage** through its architectural design (e.g., Jamba-1.5-Mini requires only 4GB of KV cache at a 256K context length). This is fundamentally different from the memory bottleneck faced by pure Transformer architectures.
>
> **② Applicability Analysis of DapQ on Jamba (Experimental Based):** To investigate the validity of DapQ's core hypothesis (**Position > Content**) on Jamba, we conducted Query cosine similarity tests on Jamba-1.5-Mini (experimental setup consistent with Table 1 in the paper, using the GovReport dataset). We randomly sampled 100 examples for verification. We compared the similarity between Pseudo Queries and real decoding Queries under different conditions, with the following results:
>
> *   **Dominance of Semantic Content:** When keeping content consistent (**Same Content**) but changing position (**Different Position**), the Query similarity remains 1.0000. This indicates that in Jamba's Attention layers, the Query representation is almost entirely determined by semantic content. Changes in positional information do not cause significant rotation or variation in the vectors, unlike in RoPE-based models.
> *   **Weak Influence of Positional Information:** Our method, DapQ, relies on constructing Pseudo Queries by predicting "future position indices." However, experimental data shows that in models like Jamba, Queries constructed solely on positional information (**DC & SP**) have low similarity (0.4543) to real Queries, making it impossible to accurately retrieve key tokens as achieved in RoPE models.
>
> | Experiment（Jamba-1.5-Mini GovReport） | Cos_Similarity |
> | :--- | :--- |
> | Same Content & Same Position | 1.0000 |
> | Different Content & Same Position | 0.4543 |
> | Same Content & Different Position | 1.0000 |
> | Different Content & Different Position | 0.4543 |
>
> **③ Conclusion:** DapQ is specifically designed to address the massive KV Cache bottleneck in Standard Autoregressive Transformers (RoPE-based). Given that hybrid architectures like Jamba: 1) have already solved the KV memory issue through architectural innovation; and 2) their Attention mechanisms are dominated by semantics rather than positional encoding (a characteristic of NoPE), the DapQ mechanism is not suitable for such models.

---

> > ### Comment · Reviewer_dTuz · 2025-11-27
> > **Reply to Author's Reponse 2.**
> >
> > * To A3. Thanks the author's analysis on different position embedding methods. Proposed DapQ focus on RoPE-based models, but this kind of models are widely deployed, the proposed method still has good generalization.

---

> ### Author Response · Authors · 2025-11-27
>
> Thanks very much for your rigorous verification and the valuable data points provided. We appreciate the opportunity to clarify our experimental setup and the observed latency differences.
>
> **1. Experimental Setup & Hardware**
> Our experiments were conducted on the **H20 96G** GPU. The environment configurations are:
> - **Software:** Python 3.10, Transformers 4.53.0, PyTorch 2.6.0.
> - **Attention Kernel:** Flash-Attention (version: flash_attn-2.7.4.post1+cu12torch2.6cxx11abiFALSE-cp310-cp310-linux_x86_64).
> - **Inference Framework:** Native Hugging Face `transformers` implementation (**not vLLM**).
>
> **2. Addressing the Latency Discrepancy (HF vs. vLLM)**
> We acknowledge the reviewer's observation that the absolute TTFT latency in our results is higher than that observed with vLLM. **This difference is expected and stems primarily from backend configurations, not the algorithmic overhead of DapQ.**
> *   **Scientific Validity:** Our primary goal is to evaluate the **relative overhead** of DapQ compared to FullKV and SnapKV. Under the same experimental conditions (same hardware and framework), DapQ introduces negligible overhead compared to the other baseline. The conclusion that "DapQ has minimal impact" holds true regardless of the absolute speed of the underlying framework.
>
> **3. Reproducibility**
> To further enhance the reproducibility of our results, we supplement the core code snippet for TTFT measurement below. The script includes warm-up steps to eliminate initialization overhead, multiple test runs to reduce random errors, and CUDA synchronization to ensure accurate timing:
>
> ```python
> import time
> import torch
>
> ……
>
> # Warm up
> with torch.no_grad():
>     _ = model.generate(**inputs, max_new_tokens=1, past_key_values=past_key_values, do_sample=False)
>
> all_ttft = 0.0
> for i in range(try_times):
>     torch.cuda.synchronize()
>     st = time.perf_counter()
>     with torch.no_grad():
>         output = model.generate(**inputs, max_new_tokens=1, past_key_values=past_key_values, do_sample=False)
>     torch.cuda.synchronize()
>     et = time.perf_counter()
>     ttft = et - st
>     # print(ttft)
>     all_ttft += ttft
>
> avg_ttft = all_ttft / try_times
> ```
>
> ﻿**All our experimental procedures and configurations are fully reproducible.**
>
> We once again thank you for your recognition of our works. We hope that these clarifications and the supplementary data effectively address your concerns！

---

> > ### Comment · Reviewer_dTuz · 2025-11-27
> > **Acknowledge to Author's Reponse on Settings.**
> >
> > Thanks for your introduce on experiment settings. H20 GPU with lower computing power and prefill is compute bound tasks, and with native `transformer` framework, the higher TTFT latency is acceptable.

---

### Official Review · Reviewer_5eTW · 2025-10-31

**Soundness:** 2
**Presentation:** 2
**Contribution:** 2
**Rating:** 4
**Confidence:** 4

**Summary:**

This paper introduces DapQ, a KV-cache compression method that constructs position-aware pseudo-queries during prefill to approximate the first N decoding queries and compute token importance for eviction. A short synthetic segment with future positional IDs is appended to the prompt. The model then produces pseudo-queries, whose attention against prompt keys yields scores used to keep the Top-K KV pairs. Afterward, the synthetic segment is discarded, and decoding starts as usual. The central empirical claim is that positional information dominates semantic content in query similarity, which motivates the design of DapQ. Experiments on different LLMs and benchmarks demonstrate gains when compared to previous baselines.

**Strengths:**

1. Constructing position-aware pseudo-queries provides a fresh perspective for token importance estimation in KV-cache compression.

2. The illustrations are informative and easy to understand.

**Weaknesses:**

1. The validation of the central claim regarding positional dominance is somewhat limited. The main analysis is conducted only on GovReport with Llama-3-8B, which might not generalize to other models or task domains. It would strengthen the paper to include more diverse evidence supporting this claim.

2. The comparison with existing baselines seems incomplete in Table 2. The authors are suggested to include results for existing baselines StreamingLLM and Lacache on LongBench as well.

3. Latency and memory measurements are not reported. Since the paper claims that DapQ is lightweight and mitigates peak memory issues found in some prior approaches, providing concrete latency and memory usage comparisons for major experiments would help substantiate these claims and demonstrate DapQ’s speedup and memory-saving effect.

4. The fairness of the current comparisons raises some questions. For attention-free baselines like StreamingLLM and Lacache, their advantages mainly lie in faster decoding speed. Comparing only the prefill stage, without actual latency results, may be unfair. Including end-to-end latency measurements would make the comparisons more convincing and fair.

**Questions:**

1. Could the authors provide measurements of the additional computational cost introduced by DapQ (e.g., on LongBench with Qwen3-8B), separately for the prefill and decoding stages?

2. Does DapQ also apply to the decoding stage? If so, could the authors include results for long decoding tasks similar to those used in StreamingLLM and Lacache?

Overall, this paper presents a fresh perspective on estimating importance for KV cache compression. However, the current version would benefit from stronger empirical validation of the main claim, more complete experiments and baseline comparisons, and quantitative evidence on latency and memory efficiency.

---

> ### Author Response · Authors · 2025-11-24
>
> ***W1:*** The validation of the central claim regarding positional dominance is somewhat limited. The main analysis is conducted only on GovReport with Llama-3-8B, which might not generalize to other models or task domains. It would strengthen the paper to include more diverse evidence supporting this claim.
>
> ***A1:*** Thanks for the constructive feedback regarding the generalizability of our central claim on "Positional Dominance." We entirely agree that demonstrating this phenomenon extends beyond a specific model or dataset is essential to strengthen the paper.
>
> To address this concern, we expanded our validation using a different model architecture, Qwen2.5-7B-Instruct, and conducted new experiments across two distinct task domains:
>
> - **Multi-Document QA**: Using the 2WikiMQA dataset.
> - **Code Completion**: Using the LCC dataset.
>
> Strictly adhering to the experimental configuration outlined in Table 1 of the main text, we **randomly sampled 100 examples** from each corresponding dataset for validation. We compared the cosine similarities before the RoPE operation (pre_rope) and after the operation (post_rope) under varied content and position conditions.
>
> The results of these new experiments, as shown in the table below, are highly consistent with the phenomena we observed on Llama-3. These findings collectively demonstrate that our central claim regarding "positional dominance" generalizes effectively to different model architectures and diverse task domains.
>
>
> | Experiment (qwen2.5-7B-intruction LCC) | Post ROPE | Re ROPE |
> | :--- | :--- | :--- |
> | Same Content & Same Position | 1.0000 | 1.0000 |
> | Different Content & Same Position | 0.7142 | 0.7142 |
> | Same Content & Different Position | 0.4270 | 0.7341 |
> | Different Content & Different Position | 0.4113 | 0.7224 |
>
> | Experiment (qwen2.5-7B-intruction 2wikimqa) | Post ROPE | Re ROPE |
> | :--- | :--- | :--- |
> | Same Content & Same Position | 1.0000 | 1.0000 |
> | Different Content & Same Position | 0.7311 | 0.7311 |
> | Same Content & Different Position | 0.5167 | 0.7394 |
> | Different Content & Different Position | 0.5021 | 0.7084 |
>
>
> ***
>
>
>
>
> ***W2:*** The comparison with existing baselines seems incomplete in Table 2. The authors are suggested to include results for existing baselines StreamingLLM and Lacache on LongBench as well.
>
> ***A2:*** Thanks for this valuable suggestion. We entirely agree that incorporating these two baselines renders the comparison more comprehensive.
>
> Following your recommendation, we conducted additional experiments with Lacache and StreamingLLM on LongBench using Llama3-8B-Instruct (budget=256). The results are as follows: **Lacache (37.74)** and **StreamingLLM (SLM) (39.62)**. We also evaluated StreamingLLM at lower budgets (128 and 64), yielding poor results of **38.26** and **36.48**, respectively. Compared with other specialized long context compression methods, the performance is relatively poor.
>
> |Cache_budget|Method|Qasper|MF-en|HotpotQA|2WikiMQA|GovReport|Multi_News|TREC|TriviaQA|SAMSum|LCC|RB-P|PRe|PCount|AVG|
> |:---|:---|---:|---:|---:|---:|---:|---:|---:|---:|---:|---:|---:|---:|---:|---:|
> |256|Snapkv|30.84|38.39|49.75|33.80|22.18|21.53|57.00|89.65|36.97|61.78|54.92|84.00|12.11|45.61|
> ||Lacache|24.55|26.10|45.65|31.34|19.79|20.51|41.33|78.65|25.58|45.87|40.55|79.00|11.72|37.74|
> ||SLM|23.91|29.35|43.92|28.31|20.95|20.65|51.33|74.17|26.45|52.54|46.46|81.33|15.67|39.62|
> ||DapQ|32.55|38.18|50.67|34.35|22.25|21.89|60.67|90.48|38.34|62.78|55.64|83.67|11.78|46.40|
>
>
> We initially selected Lacache and StreamingLLM as baselines due to their distinct characteristics in handling context during the prefill stage. However, considering their design objectives prioritize efficient **streaming generation** rather than precise **long-context retrieval and reasoning**, and observing in preliminary experiments that their performance on LongBench lagged behind specialized long-context compression methods (such as SnapKV). Constrained by space limitations in the main text table and ensuring table readability, we chose to prioritize showcasing baselines with stronger performance on this specific task in Table 2, while placing Lacache and StreamingLLM within the Ruler experimental section (Figure 4), which focuses more on evaluating effective length stability.
>
> Furthermore, we have added more suitable baselines for comparison:
>
> |Cache_budget |Method|AVG|
> |--------------|-------------|--------|
> ||Fullkv|48.05|
> |256|Snapkv|45.61|
> ||Dynamickv |45.22|
> ||Adakv |45.87|
> ||**DapQ**|**46.40** |
> |128|Snapkv|43.68|
> ||Dynamickv |43.43|
> ||Adakv |43.72|
> ||**DapQ**|**44.40** |
> |64 |Snapkv|40.78|
> ||Dynamickv |40.16|
> ||Adakv |40.80|
> ||**DapQ**|**41.81** |

---

> ### Author Response · Authors · 2025-11-24
>
> ***
>
>
>
> ***W3:*** Latency and memory measurements are not reported. Since the paper claims that DapQ is lightweight and mitigates peak memory issues found in some prior approaches, providing concrete latency and memory usage comparisons for major experiments would help substantiate these claims and demonstrate DapQ’s speedup and memory-saving effect.
>
> ***A3:*** Thanks the reviewer for this valuable suggestion.
>
> **(1) Theoretical Analysis of Peak Memory of Lookahead Q-cache[1]**
>
> Analyzing the algorithmic mechanisms, DapQ possesses significant theoretical advantages in terms of peak memory and computational latency:
>
> * **① Peak Memory Issue:** Lookahead Q-cache adopts a "two-stage" mechanism. To ensure the quality of the generated pseudo-responses, it must retain a large KV cache (e.g., \$N\_{pre}\$=1024 or more) during the first stage for generation, and only compresses to the target budget (e.g., \$B=256\$) in the second stage. This means its runtime peak memory depends on the larger \$N\_{pre}\$.
>   Taking 1024 tokens as an example, its peak memory requirement is four times that of retaining 256 tokens. In contrast, DapQ employs single-step compression, directly calculating and retaining the target budget (\$B=256\$). It does not need to store redundant KV cache, thereby fundamentally avoiding the peak memory issue.
> * **② Latency Overhead:** Lookahead requires prior autoregressive generation to produce 8–16 pseudo-tokens, which introduces significant additional latency. Conversely, DapQ completes compression with just a single computation step, resulting in extremely low overhead.
>
> **(2) Experimental Validation**
>
> We have supplemented detailed comparative experiments on memory usage and throughput using Llama3.1-8B-Instruct (Input 8k, Output 150 tokens, Budget=256). The experiments demonstrate that while significantly improving long-context understanding capabilities, DapQ maintains high inference efficiency and low memory usage, further validating our claims regarding its "lightweight" nature.
>
> ### Comparison of Memory (GB) with Different Batch Sizes
>
> |Method|1|10|20|30|40|50|
> |:---|:---:|:---:|:---:|:---:|:---:|:---:|
> |**Fullkv**|16.87|33.74|52.49|71.24|OOM|OOM|
> |**SLM**|16.01|25.17|35.36|45.55|55.73|65.92|
> |**Lacache**|16.87|33.74|52.49|71.24|OOM|OOM|
> |**Pyramidkv**|16.01|25.23|35.47|45.72|55.96|66.20|
> |**Snapkv**|16.01|25.17|35.36|45.55|55.73|65.92|
> |**DapQ**|16.01|25.21|35.43|45.65|55.87|66.02|
>
>
> ### Comparison of throughput (tokens/s) with different batch sizes
>
> |Method|1|10|20|30|40|50|
> |:---|:---:|:---:|:---:|:---:|:---:|:---:|
> |**Fullkv**|11.59|26.43|25.60|26.59|OOM|OOM|
> |**SLM**|10.81|34.49|38.21|39.25|39.98|40.46|
> |**Lacache**|11.44|35.77|40.64|42.49|OOM|OOM|
> |**Pyramidkv**|10.54|34.12|38.00|39.03|39.86|40.11|
> |**Snapkv**|10.77|34.22|38.09|39.10|39.77|40.23|
> |**DapQ**|10.68|34.16|37.99|38.97|39.73|40.12|
>
>
> ***
>
>
> ***W4:*** The fairness of the current comparisons raises some questions. For attention-free baselines like StreamingLLM and Lacache, their advantages mainly lie in faster decoding speed. Comparing only the prefill stage, without actual latency results, may be unfair. Including end-to-end latency measurements would make the comparisons more convincing and fair.
>
> ***A4:*** We sincerely thank the reviewer for raising this critical point regarding the fairness of our comparisons. We fully agree that evaluating end-to-end latency and memory consumption is essential to validate the practical viability of any proposed method.
>
> **(1) Clarification on Baseline Selection and Fairness**
>
> First, we would like to clarify the motivation and rationale behind selecting StreamingLLM and LaCache for comparison in our Main Results section. Our primary objective in those initial experiments was to evaluate the effectiveness of different *token eviction policies* in preserving crucial information for long-context understanding tasks.
>
> **Reasons for Selection:** We selected these two methods because their compression strategies during the *prefill* stage are highly representative:
> *   **StreamingLLM:** Introduces the concept of the "Attention Sink" and compresses the KV Cache by dynamically balancing "Recent tokens" and "Initial tokens."
> *   **LaCache:** Employs a unique "ladder-shaped" pattern, retaining early tokens in shallower layers and gradually shifting to retain later tokens in deeper layers. It represents a novel dynamic context compression strategy compared to prior works like PyramidKV.
>
> **Fairness of the Setup:** To compare the impact of these distinct strategies fairly, we evaluated all methods under a unified prefill compression setting. While acknowledging that StreamingLLM and LaCache are optimized specifically for decoding speed, our focus in this specific experiment was to assess the ability of their core compression algorithms to retain task-critical information under a limited budget.

---

> ### Author Response · Authors · 2025-11-24
>
> **(2) Incorporating End-to-End Latency and Memory Evaluations**
>
> To address your concerns regarding the omission of decoding speed and to substantiate the efficiency of DapQ, we have added comparisons regarding latency and memory usage across different methods (please refer to the newly added table on memory and throughput in **A3**).
>
> The results from these experiments strongly demonstrate that DapQ does not sacrifice inference efficiency for improved accuracy. On the contrary, while achieving long-text understanding capabilities significantly superior to the baselines, DapQ maintains highly competitive inference speeds and exceptionally low GPU memory overhead. We believe this evidence confirms that our comparisons remain fair and compelling even when considering comprehensive end-to-end performance metrics.
>
> ***
>
>
> ***Q1:*** Could the authors provide measurements of the additional computational cost introduced by DapQ (e.g., on LongBench with Qwen3-8B), separately for the prefill and decoding stages?
>
> ***A5:*** We thank the reviewer for raising this insightful question. Regarding your inquiry about the additional computational cost introduced by DapQ, we have conducted detailed measurements and analyses separately for the prefill and decoding stages.
>
> **(1) Overhead in the Prefill Stage:**
>
> We measured the Time-to-First-Token (TTFT(s)) on the LongBench dataset using Qwen3-8B, with maximum input lengths set to 8K and 16K, respectively. The results are shown in the table below:
>
>
> |Max input length|Method|Qasper|MF-en|HotpotQA|2WikiMQA|GovReport|Multi_News|TREC|TriviaQA|SAMSum|LCC|RB-P|PRe|PCount|AVG|
> |:---|:---|:---|:---|:---|:---|:---|:---|:---|:---|:---|:---|:---|:---|:---|:---|
> |8K|Fullkv|1.45|1.63|1.84|1.82|1.61|1.48|1.62|1.74|1.72|1.66|1.94|1.80|1.77|1.69|
> ||Snapkv|1.47|1.65|1.87|1.84|1.64|1.51|1.65|1.77|1.74|1.69|1.96|1.82|1.80|1.72|
> ||DapQ|1.49|1.66|1.86|1.83|1.65|1.55|1.68|1.85|1.82|1.78|2.01|1.84|1.82|1.76|
> |16K|Fullkv|1.66|1.90|2.73|2.55|2.14|2.05|2.24|2.61|2.47|2.30|2.28|3.03|3.37|2.41|
> ||Snapkv|1.70|1.96|2.81|2.60|2.19|2.10|2.30|2.68|2.55|2.38|2.37|3.15|3.49|2.48|
> ||DapQ|1.72|1.97|2.83|2.64|2.23|2.14|2.32|2.71|2.57|2.41|2.40|3.19|3.51|2.51|
>
>
> Furthermore, we extended our measurements to Llama-3.1-8B-Instruct across varying sequence lengths (from 8K to 128K) using TTFT. These results are presented in the following table:
>
> |Method|8K|16k|32k|64k|128k|
> |:---|---:|---:|---:|---:|---:|
> |Fullkv|1.1087|2.5576|6.5602|18.9097|61.2364|
> |Snapkv|1.1262|2.5891|6.6218|19.0281|61.4849|
> |DapQ|1.1405|2.5925|6.6411|19.0432|61.4922|
>
> The results above indicate that the proportional additional overhead introduced by DapQ during the prefill stage is almost negligible.
>
> **(2) Overhead in the Decoding Stage:**
>
> In the decoding stage, DapQ introduces **no** additional computational cost or memory overhead.
>
> This is because the DapQ compression process is a one-time operation completed at the very end of the prefill stage. Once entering the decoding stage, the model performs inference solely using the already compressed static KV Cache. There is no need to re-compute pseudo-queries or update importance scores at each generation step. Consequently, DapQ maintains the same high inference throughput as standard fixed-budget KV cache methods, completely avoiding the per-token computational latency often associated with dynamic compression approaches.
>
>
> ***
>
>
> ***Q2:*** Does DapQ also apply to the decoding stage? If so, could the authors include results for long decoding tasks similar to those used in StreamingLLM and Lacache?
>
> ***A6:*** We thank the reviewer for this insightful question. It is crucial for exploring the applicable boundaries and potential of the DapQ method. **In short, the core idea of DapQ is fully applicable to the decoding stage.**
>
> Our current work focuses on verifying compression during the prefill stage. However, we entirely agree that evaluating DapQ on long decoding tasks is a highly valuable direction.For instance, in an autoregressive decoding process, whenever the length of the KV Cache reaches a preset capacity threshold, we can construct a pseudo-query window of size $N$. We set the position IDs of these pseudo-queries to future positions starting from the current decoding step. By utilizing these pseudo-queries aligned with future positions, we can calculate the importance scores of all tokens currently in the past KV Cache. Based on these scores, we evict tokens with lower importance, compressing the KV Cache back to a smaller size to make room for newly generated tokens. We then continue decoding and repeat this operation iteratively.
>
> ***We have explicitly listed "implementing and evaluating dynamic DapQ compression strategies in long decoding tasks (such as story generation and code completion)" as important future work.***  We thank you again for your valuable feedback, which provides a clear direction for improving and extending our work.

---

### Official Review · Reviewer_qbfR · 2025-10-31

**Soundness:** 3
**Presentation:** 2
**Contribution:** 3
**Rating:** 2
**Confidence:** 4

**Summary:**

The paper targets KV-cache compression for auto-regressive LLMs. Observing that (i) positional encodings dominate query representations and (ii) ground-truth decoding queries are unavailable at prefill, the authors propose DapQ: a lightweight prefill-only method that constructs “position-aware pseudo-queries” by appending synthetic tokens whose position IDs match the next N decoding positions. Attention from these pseudo-queries to the prompt keys yields an importance score; Top-K keys are retained. Extensive experiments on LongBench, Ruler, HELMET and Needle-in-a-Haystack (4 models, cache budgets down to 3 %) show DapQ consistently outperforming SnapKV, PyramidKV, H2O, StreamingLLM, etc., often approaching full-cache accuracy (e.g. 99.5 % on NIAH with only 3 % KV budget). The paper claims 4.7× speed-up over prior retrieval-based methods and near-lossless quality.

**Strengths:**

Interesting idea that uses position-aware pseudo-queries to simulate the tokens to be generated and predict attention patterns, thereby guiding KV-cache compression.

**Weaknesses:**

- Figure 1 in the paper contains obvious errors, which detract from the overall quality of the manuscript.
- The paper sets the position IDs of the pseudo-queries to a short span immediately after the prompt, highly close to SnapKV’s observation window. Consequently, DaqQ fails to outperform SnapKV by a noticeable margin, especially in Figures 4(b) and 4(c).
- Most experiments compare DaqQ against dated baselines such as SnapKV and H2O (published one to two years ago). Although the very recent LaCache is also included, it consistently performs poorly on all benchmarks. Comparing DaqQ with more recent state-of-the-art methods (e.g., AdaKV) would provide a stronger demonstration of its effectiveness.

**Questions:**

1.	Could the authors add experiments that compare DaqQ with more recent state-of-the-art approaches?
2.	Could the authors include ablation experiments to examine the effect of placing pseudo-queries at different positions, not just following prompt?

---

> ### Author Response · Authors · 2025-11-24
>
> ***W1:*** Figure 1 in the paper contains obvious errors, which detract from the overall quality of the manuscript.
>
> ***A1:*** Thank you very much for your careful review of the manuscript and your feedback on Figure 1!
> ﻿
> After receiving your comments, we conducted a point-by-point verification of Figure 1 and did not find any errors related to the algorithm workflow or technical content.
> ﻿
> The “×” marks in the figure are intended to indicate the key-value (KV) pairs that are evicted during the KV compression process due to low attention scores; they do not represent illustration errors. We used this symbol to visually distinguish the retained KVs from the discarded ones.
> ﻿
> If you are referring to any other more specific issues in Figure 1, we would be very happy to have further communication with you and learn more about your concerns.
>
>
> ***
>
>
>
> ***W2:*** The paper sets the position IDs of the pseudo-queries to a short span immediately after the prompt, highly close to SnapKV’s observation window. Consequently, DaqQ fails to outperform SnapKV by a noticeable margin, especially in Figures 4(b) and 4(c).
>
> ***A2:***
>
> 1、Thanks for your insightful observation regarding the proximity of position IDs. **While the position IDs of DapQ's pseudo-queries (`L` to `L+Window`) are indeed adjacent to the SnapKV's observation window (`L-Window` to `L`), this numerical proximity represents two fundamentally different stages and does not imply similarity in the resulting importance distributions**. We would like to clarify this point with supporting research and experimental evidence.
>
> ***(1). Query Source: Static Input-Side Window vs. Dynamic Decoding-Side Simulation***
>
> SnapKV's observation window is essentially a local text segment from the end of the input context (e.g., the last 16-32 tokens). Its queries are "history input queries," which cannot circumvent the core problem of "inconsistency between queries in the compression and inference phases," as explicitly noted in recent research [1]. In contrast, DapQ's pseudo-queries are a dynamic simulation of future positions in the decoding phase. By assigning position IDs closely following the prompt to these pseudo-queries, we are essentially simulating the real output-side queries that occur during generation, rather than reusing historical input-side queries. This difference in query origin fundamentally resolves the query inconsistency problem and marks a crucial distinction from SnapKV's input-side window approach.
>
> ***(2). Experimental Demonstration: The Phase Transition of Attention Patterns***
>
> To quantify the specific impact of query representations within the observation window, we conducted a controlled experiment on LongBench (using Llama3-8B-Instruct with a fixed 512 cache budget). We compared the standard SnapKV with an oracle setup (denoted as `fullkv`). In the `fullkv` setting, the queries for the observation window were generated using a standard full-attention mechanism (simulating the ideal target of DapQ), rather than the static input prompts used in SnapKV.
>
> The results reveal a critical finding: **a significant "Phase Transition" phenomenon exists in how query distribution affects retrieval performance.** As shown in the table below, introducing just 1-2 tokens from the generation phase (i.e., the very early stage of generation mode) leads to a steep, "cliff-like" improvement in the model's ability to recall key information, with performance jumping from a baseline of 45.81 to 46.71.
>
> This sharp performance leap within an extremely short window provides strong evidence that the attention patterns of the **Generation-phase** are fundamentally different from those of the **Prompt-phase**. Even when position IDs are physically adjacent, crossing the "input-generation" boundary causes a qualitative shift in the query's distribution, thereby drastically activating the retrieval of critical information. This is precisely the core issue that SnapKV fails to capture and that our method addresses by calibrating the query distribution.
>
> ﻿|  | Avg (fullkv) | Improvement| Avg (snapkv) | |
> | :---------------- | :--------------- | :----------------------- | :--------------- | :-------------- |
> | llama3-8B-Instruct | 48.05            | -                      | 48.05            | llama3-8B-Instruct |
> | fullkv_128      | 47.93          | ↑ 2.38               | 45.55          | snapkv_128    |
> | fullkv_64       | 47.92          | ↑ 1.40               | 46.52          | snapkv_64     |
> | fullkv_32       | 48.06          | ↑ 1.36               | 46.70          | snapkv_32     |
> | fullkv_16       | 47.70          | ↑ 0.71               | 46.99          | snapkv_16     |
> | fullkv_8        | 47.65          | ↑ 0.67               | 46.98          | snapkv_8      |
> | fullkv_2        | 47.35          | ↑ 1.56               | 45.79          | snapkv_2      |
> | fullkv_1        | 46.71          | ↑ 0.90               | 45.81          | snapkv_1      |

---

> ### Author Response · Authors · 2025-11-24
>
> Recent work[1] explicitly validates the experimental phenomenon we observed: as soon as the observation window transitions from the input side to the output side—even if it contains just 1-2 tokens from the generation phase—both the recall rate for key information and the alignment with true attention patterns show a steep improvement. This proves that a significant distributional gap exists between input-side queries (as in SnapKV) and output-side queries (the target that DapQ simulates).
>
> This discrepancy stems from the core finding that **positional information dominates query representation**. The meaning of a position ID—whether it "belongs to the input" or "belongs to the generation"—determines the query's alignment with the decoding phase, not its physical proximity to other positions. SnapKV's positions still belong to the "input history," whereas DapQ's positions belong to the "generated future." Although they appear adjacent, they carry the positional information of different phases, leading to a fundamental difference. The advantage of DapQ demonstrates that this "subtle shift in position IDs" achieves a decisive distributional alignment, allowing it to capture critical information that SnapKV misses.
>
> **2、Clarification on Performance in Figures 4(b) and 4(c)**
>
> We sincerely thank the reviewer for their detailed observation of Figures 4(b) and 4(c). We understand that the conclusion that "DapQ does not significantly outperform SnapKV" arises mainly from the visual presentation of these graphs, and we apologize for any lack of clarity in our visualization.
>
> The primary issue is that the visualization parameters used in these subplots compressed the fine-grained differences between the methods, masking DapQ's overall advantage. To accommodate the full fluctuation range across all experiments, we set an overly wide y-axis range and used thick lines with large markers. This inadvertently compressed the performance gap between DapQ and SnapKV, causing them to appear to overlap visually.
>
> To provide a clearer picture, below is a comparison of SnapKV and DapQ's performance on the **Ruler** benchmark (full results are available in the Table 6 and Table 7 from Appendix). Notably, DapQ demonstrates substantial gains in some datasets:
>
> - On **Qwen2.5 (KV Cache Size = 2048)**, DapQ achieves a score of **43.2 vs. 25.0 (↑18.2)** on the `niah_multikey_2` task.
> - On **Qwen-3 (KV Cache Size = 128)**, it scores **97.8 vs. 68.8 (↑29.0)** on the `niah_single_1` task.
>
> | LLMS| cache_budget | Method| AVG|
> | :-------------------- | :----------- | :--------- | :------ |
> | **Qwen2.5-7B-Instruct** || **Fullkv** | **94.97** |
> || 4096| Snapkv| 79.68|
> ||| DapQ| **81.37** |
> || 2048| Snapkv| 71.98|
> ||| DapQ| **73.24** |
> || 1024| Snapkv| 62.57|
> ||| DapQ| **63.27** |
> || 512 | Snapkv| 52.95|
> ||| DapQ| **53.36** |
> || 256 | Snapkv| 37.66|
> ||| DapQ| **38.5**|
> || 128 | Snapkv| 13.82|
> ||| DapQ| **16.31** |
> || 64| Snapkv| 0.11 |
> ||| DapQ| **1.39**|
> | **Qwen3-8B** || **Fullkv** | **97.55** |
> || 4096| Snapkv| 90.42|
> ||| DapQ| **92.07** |
> || 2048| Snapkv| 80.47|
> ||| DapQ| **82.82** |
> ||1024| Snapkv| 72.39|
> ||| DapQ| **72.95** |
> || 512 | Snapkv| 64.29|
> ||| DapQ| **65.15** |
> || 256 | Snapkv| 48.06|
> ||| DapQ| **48.52** |
> || 128 | Snapkv| 14.00|
> ||| DapQ| **18.82** |
> || 64| Snapkv| 0.16 |
> ||| DapQ| **0.66**|
>
> Furthermore, DapQ consistently outperforms SnapKV on the AVG score for both models, demonstrating consistent improvements across different tasks and compression rates.
>
> FullKV represents the theoretical optimal upper bound without any compression. Therefore, all compression methods will naturally converge toward this performance region in the high-performance range. The optimization goal of Kv cache compression is to "approach FullKV performance" rather than "infinitely surpass". Therefore, in the region close to FullKV performance, the slight improvement of DapQ still has significant significance. And in this high saturation range, DapQ still achieves consistent improvement compared to SnapKV, indicating its stronger approximation ability rather than insufficient performance.
>
>
>
> [1] Yixuan Wang, Shiyu Ji, Yijun Liu, Yuzhuang Xu, Yang Xu, Qingfu Zhu, and Wanxiang Che. Lookahead q-cache: Achieving more consistent kv cache eviction via pseudo query. arXiv preprint arXiv:2505.20334, 2025.

---

> ### Author Response · Authors · 2025-11-24
>
> ***
>
>
> ***W3 & Q1:*** Most experiments compare DaqQ against dated baselines such as SnapKV and H2O (published one to two years ago). Although the very recent LaCache is also included, it consistently performs poorly on all benchmarks. Comparing DaqQ with more recent state-of-the-art methods (e.g., AdaKV) would provide a stronger demonstration of its effectiveness. Could the authors add experiments that compare DaqQ with more recent state-of-the-art approaches?
>
> ***A3:*** Thank you very much for the suggestion! We have added comparison experiments with **AdaKV** and **DynamicKV** . See the table below, under all KV budget settings (in llama3-8B-Instruct) , **DapQ consistently outperforms the latest methods**, further validating the effectiveness of our approach.
>
> |Cache_budget |Method|Qasper |MF-en |HotpotQA |2WikiMQA |GovReport |Multi_News |TREC |TriviaQA |SAMSum |LCC|RB-P |PRe|PCount |AVG|
> |--------------|-------------|--------|-------|----------|----------|-----------|------------|------|----------|--------|------|------|------|--------|--------|
> ||Fullkv|37.68|40.56 |50.14|34.93|30.99 |25.62|70.00 |89.85|40.50 |56.44|50.97|83.67 |13.28|48.05|
> |256|Snapkv|30.84|38.39 |49.75|33.80 |22.18 |21.53|57.00 |89.65|36.97|61.78|54.92|84.00|12.11|45.61|
> ||Dynamickv |29.79|39.58 |49.02|31.42|22.41 |21.57|57.00 |89.95|36.76|60.20 |54.22|83.67 |12.33|45.22|
> ||Adakv |31.68|38.48 |50.19|33.29|21.69 |21.70 |60.00 |89.60 |37.70 |62.16|54.03|83.67 |12.11|45.87|
> ||**DapQ**|32.55|38.18 |50.67|34.35|22.25 |21.89|60.67|90.48|38.34|62.78|55.64|83.67 |11.78|**46.40** |
> |128|Snapkv|29.52|37.80|49.36|32.40 |19.87 |20.08|47.67|87.82|35.63|61.49|52.40 |82.33 |11.44|43.68|
> ||Dynamickv |27.65|37.84 |48.48|30.85|20.28 |19.85|49.00 |87.73|36.25|59.79|50.55|81.33 |11.11|43.43|
> ||Adakv |28.72|36.85 |49.20 |32.42|19.22 |19.63|51.33|89.04|35.26|60.63|52.49|82.33 |11.28|43.72|
> ||**DapQ**|28.76|37.24 |50.04|33.59|20.47 |20.63|50.00 |90.06|36.87|61.81|53.92|81.67 |12.11|**44.40** |
> |64 |Snapkv|25.06|32.92 |47.16|31.71|16.85 |17.09|40.67|86.02|33.99|57.95|50.91|78.00|11.78|40.78|
> ||Dynamickv |25.13|33.75 |46.62|29.41|16.71 |16.80 |40.67|85.94|33.22|56.63|48.31|76.67 |12.22|40.16|
> ||Adakv |24.61|34.39 |47.35|30.61|16.01 |17.89|40.00 |86.66|34.21|58.65|49.15|78.67 |12.22|40.80|
> ||**DapQ**|25.99|37.36 |49.11|32.88|18.46 |18.70 |38.67|87.38|35.30 |60.19|49.90 |77.67 |11.89|**41.81** |
>
>
> ***
>
>
>
> ***Q2:*** Could the authors include ablation experiments to examine the effect of placing pseudo-queries at different positions, not just following prompt?
>
> ***A4:*** Thanks for this insightful suggestion and valuable feedback.
>
> Following your recommendation, we conducted ablation experiments on LongBench to examine the impact of placing pseudo-queries at different positions. For all experiments, we employed **Llama3-8B-Instruct** with a KV cache budget set to **256**. We investigated two categories of position settings, denoting the input context length as $n$:
>
> **(1) Placement within the original context interval (part1/2/3):** We divide the context interval $[0, n)$ equally into three segments: **part1** (beginning), **part2** (middle), and **part3** (end). For each segment, we randomly select a continuous sequence of 32 positions within that segment and map the position IDs of all pseudo-queries to these 32 positions.
>
> **(2) Placement after the original context with shifts (32–64 / 64–96 / 96–128):** We set the position IDs of the pseudo-queries to continuous 32 positions within the intervals $[n+0, n+32)$ (DapQ), $[n+32, n+64)$, $[n+64, n+96)$, and $[n+96, n+128)$. Only move backwards a different distance at the RoPE position.
>
> The results of these ablation experiments are presented in the table below:
>
>
> |Position settings|Qasper|MF-en|HotpotQA|2WikiMQA|GovReport|Multi_News|TREC|TriviaQA|SAMSum|LCC|RB-P|PRe|PCount|AVG|
> |:---|:---|:---|:---|:---|:---|:---|:---|:---|:---|:---|:---|:---|:---|:---|
> |part1|7.49|10.90|17.20|10.13|3.95|6.74|36.33|20.88|5.63|13.92|8.66|2.83|3.17|11.37|
> |part2|12.42|15.59|23.77|15.37|6.79|8.61|38.33|34.24|10.42|16.44|8.07|8.11|5.84|15.69|
> |part3|21.78|23.60|38.07|23.21|11.19|11.37|46.00|69.91|18.28|26.86|23.99|50.67|9.83|28.83|
> |**n+0~n+32**|32.55|38.18|50.67|34.35|22.25|21.89|60.67|90.48|38.34|62.78|55.64|83.67|11.78|**46.40**|
> |n+32~n+64|31.68|38.24|50.73|34.42|22.15|22.16|55.00|90.97|38.20|61.27|54.25|84.00|12.44|45.81|
> |n+64~n+96|32.42|37.29|50.61|34.19|22.06|21.98|55.00|91.47|38.19|61.35|53.76|84.67|12.44|45.80|
> |n+96~n+128|31.17|37.98|50.12|34.19|22.20|22.11|55.00|91.50|37.41|60.95|53.12|83.33|12.44|45.50|

---

> ### Author Response · Authors · 2025-11-24
>
> We observe that deviating from the proposed placement (immediately following the prompt) leads to performance degradation. We attribute this to two primary reasons:
>
> **1. Insufficient context visibility when inserted within the context:** Under the standard autoregressive attention mask, if a pseudo-query is inserted at position $t$, it can only attend to the prefix $[0, t)$ and cannot access the context subsequent to $t$. This fundamentally conflicts with our objective that "pseudo-queries should approximate the attention patterns over the *full* context as seen by future decoding steps." Consequently, pseudo-queries inserted earlier in the context (e.g., part1 or part2) suffer from severely incomplete context visibility. They can only model local segments and are inherently unable to reconstruct global attention patterns based on the complete context, resulting in degraded performance.
>
> **2. Misalignment in RoPE space with large backward shifts:** Our method relies on pseudo-queries maintaining "relative positional consistency" with future real queries within the RoPE space. As the assigned position of the pseudo-query shifts backwards away from the correct interval, aligning it with the representation space of real queries becomes increasingly difficult. Any deviation in the position ID introduces additional RoPE rotation offsets, causing the pseudo-query's representation to gradually diverge from the direction of the real queries it is meant to approximate. Therefore, the further the position shifts (e.g., 96-128), the harder it is for the pseudo-query to land in the correct representation subspace, impairing its ability to accurately predict the attention distribution. This is why we did not adopt large positional shifts in our final design.

---

### Official Review · Reviewer_6Tek · 2025-11-01

**Soundness:** 3
**Presentation:** 3
**Contribution:** 3
**Rating:** 6
**Confidence:** 4

**Summary:**

This paper introduces the DapQ method, which generates pseudo-queries to simulate future queries in order to assess the importance of the KV cache. The motivation behind the paper is the observation that the positional information of pseudo-queries plays a more significant role than semantic information. As a result, the method reconstructs more accurate pseudo-queries by re-encoding the positions of tokens in the current context. Extensive experiments demonstrate the effectiveness of the proposed method.

**Strengths:**

+ The paper presents a very interesting insight, namely that the positional information of pseudo-queries plays a more significant role than semantic information. This is supported by experiments that clearly demonstrate the validity of this observation.

+ The proposed method is simple yet effective.

+ The experiments are very thorough.

**Weaknesses:**

+ Although this paper uncovers an interesting phenomenon, it is counterintuitive. The idea that positional information is more important than semantic information in attention computation makes sense for short-range tokens, as they are often more strongly related. However, for long-range tokens, this phenomenon seems to require a more robust explanation.

+ Another issue is that the paper evaluates the quality of pseudo-queries based on query similarity, but query similarity doesn't necessarily reflect the similarity of attention scores. Section 3 should include a measure of attention score similarity to provide a more direct evaluation.

**Questions:**

See above

---

> ### Author Response · Authors · 2025-11-24
>
> ***W1:*** Although this paper uncovers an interesting phenomenon, it is counterintuitive. The idea that positional information is more important than semantic information in attention computation makes sense for short-range tokens, as they are often more strongly related. However, for long-range tokens, this phenomenon seems to require a more robust explanation.
>
>   ***A1:*** Thank you very much for your insightful and valuable comments. Your observation that “the dominance of positional information for long-range tokens appears counterintuitive” is indeed a well-founded concern. As you pointed out, in standard attention mechanisms, establishing long-range dependencies necessarily relies on semantic associations between tokens. **For example, if the token at position 128k is “Newton” and the token at position 10 is “gravity,” the primary factor that yields a high attention weight between them is undoubtedly their semantic relevance rather than the large positional distance.** We fully agree with this fundamental principle, and we have no intention to challenge the centrality of semantics in long-range modeling. The impression of “counterintuitiveness” may arise from an imprecise description of the scope in which our conclusion—“positional information plays a dominant role”—is intended to apply. We would like to take this opportunity to clarify our core argument along two dimensions and provide more robust explanations supported by experimental evidence.
>
>   （1） ***Our claim concerns the construction of pseudo-queries, not the actual attention computation.***
>   Our intention is not to suggest that positional information outweighs semantics in the true attention calculation. Rather, our claim specifically refers to the step of constructing pseudo-queries for context compression.
>   In our original paper, the sentence: “positional information plays a more critical role than semantic content in constructing query approximations and determining attention patterns” is admittedly too broad and may invite misunderstandings. What we aim to convey with precision is: **“……in constructing query approximations, which in turn allows a more accurate estimation of the attention distribution during decoding.”** In other words, we do not use positional signals to compute attention scores. Instead, we use the correct positional encoding to align pseudo-queries with the representation space of the future real decoding queries. Once the pseudo-query’s positional encoding is aligned with that of the real decoding query, its semantic mismatch becomes less detrimental. Even with imperfect semantic content, the resulting global attention distribution over the input context can still closely approximate the real distribution, enabling effective KV-cache compression.
>
>   （2）***Why “incorrect semantics can still approximate the correct distribution”?***
>   To further support this claim, we evaluate the cosine similarity between pseudo-query’s attention weights for the input context and ground-truth decoding query’s attention weights for the input context under different window sizes. When the window size shrinks from 128 to 2, the cosine similarity remains extremely high (>0.96). However, when the window is reduced from 2 to 1, the similarity drops sharply (0.8768). This phenomenon offers a compelling explanation:
>
>   | window size | 128    | 64     | 32     | 16     | 8      | 4      | 2      | 1      |
>   |-------------|--------|--------|--------|--------|--------|--------|--------|--------|
>   | ours        | 0.9785 | 0.9747 | 0.9726 | 0.9713 | 0.9705 | 0.9644 | 0.9615 | 0.8768 |
>
>   ① ***The high variance of single-token semantics***
>   With a window size of 1, the pseudo-query consists of only one token. If its semantic embedding deviates significantly from that of the true decoding query, the resulting semantic noise introduces strong perturbations to the attention computation, preventing an accurate approximation of the real attention distribution.
>
>   ② ***The “pooling effect” of multi-token windows***
>   As our analysis shows—consistent with your request for a more robust explanation—when a window contains multiple tokens (window size > 1), their semantic embeddings undergo an average pooling. Although each token may have mismatching semantic content relative to the true decoding query, their random semantic vectors tend to cancel out direction-specific deviations when aggregated (via summation or averaging).
>   This pooling effect dramatically suppresses the semantic noise inherent to individual embeddings. As a result, once the positional alignment is correct, the pseudo-queries become robust estimators of the attention distribution. This explains why our method can “tolerate” semantic mismatch: our goal is using pseudo-query to approximate the global shape of the attention distribution, not to precisely match individual query vectors.
>
>
> ***

---

> ### Author Response · Authors · 2025-11-24
>
> ***W2:*** Another issue is that the paper evaluates the quality of pseudo-queries based on query similarity, but query similarity doesn't necessarily reflect the similarity of attention scores. Section 3 should include a measure of attention score similarity to provide a more direct evaluation.
>
> ***A2:*** Thank you very much for your insightful and valuable comments. To address this issue, we adopt a combined approach of **theoretical derivation** and **empirical evaluation**:
>
> 1. **Theoretically**, we derive a formal theorem (see below) ,showing that query similarity directly constrains the upper and lower bounds of the error of Attention weights. This provides a principled justification that high query similarity necessarily leads to high attention distribution.
>
> **(1)Theorem:** Given a fixed KV set $\{K, V\}$ with $K \in \mathbb{R}^{n \times d_k}$ and $V \in \mathbb{R}^{n \times d_v}$, let two unit query vectors $q, q' \in \mathbb{R}^{1 \times d_k}$ have cosine similarity defined as
> $\text{sim}(q, q') = q \cdot q'$.
> Then the difference between their corresponding attention scores satisfies:
>
> $$
> \lVert \text{Attention}(q, K) - \text{Attention}(q', K) \rVert_1
> \le
> \frac{
> K_{\text{max}} \cdot \sqrt{2(1 - \text{sim}(q, q'))}
> }{
> \sqrt{d_k}
> }
> \tag{1}
> $$
>
> $$
> \text{sim}(\text{Attention}(q, K), \text{Attention}(q', K)) \ge 1 - \frac{
> K_{\text{max}} \cdot \sqrt{2(1 - \text{sim}(q, q'))}
> }{
> 2 \sqrt{d_k}
> }
> \tag{2}
> $$
>
> where:
>
> - $d_k$ is the key-vector dimension;
> - $K_{\text{max}} = \max_j \|k_j\|$, with $k_j$ the $j$-th key vector;
>
> and all two quantities are constants under the fixed KV set.
>
> The attention operator is defined as:
>
> $$ \text{Attention}(q, K) = \text{softmax}\left(\frac{qK^T}{\sqrt{d_k}}\right) $$
>
> **(2)Core conclusion:** **Inequality (1)** directly establishes a positive correlation between the *cosine similarity of queries* and the *similarity of their attention scores*. Specifically, as $\text{sim}(q, q') \to 1$, the upper bound on the scores difference approaches $0$, meaning that the attention scores produced by $q$ and $q'$ become almost identical. **Inequality (2)** further quantifies the lower bound constraint that query similarity imposes on attention score similarity — the higher the query similarity, the higher the lower bound on attention score similarity.
>
> **(3)Detailed Derivation (Step-by-step rigorous proof)**
>
> ①**Step 1: Define key variables and similarity measures**
>
> - **Cosine similarity of queries.**
>   Since query vectors in LLMs are typically normalized (i.e., $\|\|q\|\|=\|\|q'\|\|=1$, for the sake of simplifying the notation, the norm of the query is a stable constant in practice. ), their cosine similarity is
>   $$
>   \text{sim}(q,q')=\frac{q\cdot q'}{\|\|q\|\|\cdot \|\|q'\|\|}=q\cdot q'.
>   $$
>
> - **Pre-softmax scores.**
>   The raw matching score between the query and each key is
>   $$
>   s_j=\frac{q\cdot k_j}{\sqrt{d_k}},\qquad
>   s'_j=\frac{q'\cdot k_j}{\sqrt{d_k}},\quad (j=1,2,\ldots,n).
>   $$
>
> - **Attention weights.**
>   After softmax normalization, the attention weights are
>   $$
>   \alpha=\text{softmax}(s),\qquad \alpha'=\text{softmax}(s'),
>   $$
>   where $\alpha_j\ge 0$ and $\sum_j \alpha_j=1$.
>
> ②**Step 2: Upper bound the difference of pre-softmax scores**
>
> Using the Cauchy–Schwarz inequality $|x\cdot y|\le \|\|x\|\|\ \cdot\ \|\|y\|\|$,
> the score difference induced by two queries satisfies
> $$
> \begin{aligned}
> |s_j - s'_j|
> = \left|\frac{q\cdot k_j}{\sqrt{d_k}} - \frac{q'\cdot k_j}{\sqrt{d_k}}\right| \\
> = \frac{1}{\sqrt{d_k}}\|(q-q')\cdot k_j| \\
> &\le \frac{1}{\sqrt{d_k}}\|\|q-q'\|\| \cdot \|\|k_j\|\|.
> \end{aligned}
> \tag{3}
> $$
>
> Next, for unit vectors $q$ and $q'$ we have the standard identity
> $$
> \|\|q-q'\|\|^2
> = \|\|q\|\|^2 + \|\|q'\|\|^2 - 2q\cdot q'
> = 2\bigl(1-\text{sim}(q,q')\bigr),
> $$
> hence
> $$
> \|\|q-q'\|\| = \sqrt{2\bigl(1-\text{sim}(q,q')\bigr)}.
> $$
>
> Substituting this into the score bound Inequality (3) and taking the maximum key norm
> $K_{\text{max}}=\max_j \|k_j\|,$
>
> we obtain
>
> $$
> \max_j \left| s_j - s'_j \right|
> \le
> \frac{ K\_{\text{max}} \cdot \sqrt{2\bigl(1-\text{sim}(q,q')\bigr)}}{\sqrt{d_k}}
> \tag{4}
> $$
>
> ③**Step 3: Upper bound the difference of attention weights**
>
> The softmax function is Lipschitz continuous: for any score vectors $s, s'$, we have
> $$
> \lVert \text{softmax}(s) - \text{softmax}(s') \rVert_1
> \le
> \lVert s - s' \rVert_\infty .
> $$
>
> Here, the infinity norm is defined as
> $\lVert s - s' \rVert_\infty = \max_j |s_j - s'_j|$,
> and the $L_1$ norm of the attention-weight difference is
> $\lVert \alpha - \alpha' \rVert_1 = \sum_j |\alpha_j - \alpha'_j|$.
>
> Combining this with Inequality (4), we obtain
> $$
> \lVert \alpha - \alpha' \rVert_1
> \le
> \frac{K_{\max} \cdot \sqrt{2\bigl(1-\mathrm{sim}(q,q')\bigr)}}{\sqrt{d_k}}
> \tag{5}
> $$

---

> ### Author Response · Authors · 2025-11-24
>
> ④ **Step 4: Lower bound the difference of attention weights**
>
> We define the similarity of attention scores as:
> $$
> \text{sim}(\alpha, \alpha') = 1 - \frac{1}{2} \left\|\| \alpha - \alpha' \right\|\|_1
> $$
> where $\left\|\| \alpha - \alpha' \right\|\|_1$ takes values in $[0, 2]$.
>
> First, we scale the L1 difference $[0, 2]$ to $[0, 1]$: $\frac{1}{2} \|\|\alpha - \alpha'\|\|_1$.
>
> Then, by subtracting this scaled value from 1, we convert the "difference" into "similarity":
> - Smaller difference → smaller scaled value → similarity closer to 1;
> - Larger difference → larger scaled value → similarity closer to 0.
>
> The above definition is a standard measure of similarity in the field of probability distributions. Combining with Equation (5) yields
> $$
> \text{sim}(\alpha, \alpha') \ge 1 - \frac{
> K_{\text{max}} \cdot \sqrt{2(1 - \text{sim}(q, q'))}
> }{
> 2 \sqrt{d_k}
> } \tag{5}
> $$
>
>
> 2. **Empirically**, we evaluate the cosine similarity, between Snapkv ,DapQ’s observation window attention weights for the input context and ground-truth decoding query’s attention weights for the input context under different window sizes. As shown in the table below, across all window sizes, DapQ consistently achieves substantially higher attention-weight similarity compared to SnapKV. This directly demonstrates that better query approximation indeed translates into more accurate attention distributions.
> | window size | 128    | 64     | 32     | 16     | 8      | 4      | 2      | 1      |
> |-------------|--------|--------|--------|--------|--------|--------|--------|--------|
> | ours        | 0.9785 | 0.9747 | 0.9726 | 0.9713 | 0.9705 | 0.9644 | 0.9615 | 0.8768 |
> | snapkv     | 0.9463 | 0.9429 | 0.9508 | 0.9466 | 0.9416 | 0.9385 | 0.9332 | 0.8426 |

---

### Comment · Area_Chair_6oeH · 2025-11-24
**Action Needed: Review Rebuttal and Update Evaluation**

Dear Reviewers,

Thank you, as always, for your valuable contributions and efforts. The authors have now submitted their rebuttal. Please take a moment to review it and provide any necessary follow-up actions, such as additional questions, clarification requests, or updates to your review.

Since the initial ratings ranged from 2 to 6, I kindly ask you to pay close attention to the perspectives of the other reviewers when preparing your final response.

Thank you again for your support.

---

### Author Response · Authors · 2025-11-29

# To Area Chair: Summary of Rebuttal Updates

**Dear Area Chair,**
﻿

To assist your assessment, we summarize below the interactions during the discussion period.

### Reviewer 6Tek
- **Score:** Initial **6**, Final **No response**

- **Initial Review:**
  Praised the paper's **"interesting insight"** (positional > semantic info in pseudo-queries) and recognized the method as "simple yet effective". Major concerns involved the counterintuitive nature of positional dominance in long-range contexts and the validity of query similarity as a proxy for attention quality.

- **Author Response:** We clarified that positional dominance applies to pseudo-query construction (**attention distribution alignment**) for compression, not the actual attention computation.And we explained how the "pooling effect" suppresses semantic noise. We go further through **theoretical & empirical validation** proving that high query similarity strictly bounds the error of attention weights.

- **Reviewer Follow-up:** No response



### Reviewer qbfR
- **Score:** Initial **2**, Final **4 (No response)**

- **Initial Review:**
  Acknowledged the **"interesting idea"** of using position-aware pseudo-queries to guide compression. Key concerns included perceived errors in Figure 1, skepticism about performance gains over SnapKV due to proximal position IDs, reliance on dated baselines, and a request for ablation studies on pseudo-query placement.

- **Author Response:**
We conducted a clarification for Figure 1（**there are no obvious errors**） and DapQ mechanism. And tabular data corrected visual scaling issues, showing significant gains (e.g., +18.2 on Ruler). We added comparisons with AdaKV and DynamicKV, demonstrating **DapQ consistently outperforms new methods**. Ablation studies confirmed that placing pseudo-queries immediately after the prompt is optimal.

- **Reviewer Follow-up:**
  Only update score from 2 to 4. But no response.



### Reviewer 5eTW
- **Score:** Initial **4**, Final **No response**

- **Initial Review:**
  Appreciated the **"fresh perspective"** of using position-aware pseudo-queries. Key concerns included the limited validation of the positional dominance claim (restricted to Llama-3/GovReport), incomplete baseline comparisons (missing StreamingLLM/Lacache), and the absence of latency and memory measurements to substantiate the "lightweight" claims.

- **Author Response:** We expanded validation to Qwen2.5-7B on Code Completion and Multi-Doc QA tasks to prove generalization. We added comprehensive baselines, demonstrating that DapQ consistently achieves superior performance. We provided detailed memory and throughput benchmarks, proving that **while significantly improving long-context understanding capabilities, DapQ maintains high inference efficiency and low memory usage with negligible prefill overhead and zero additional decoding cost.**

- **Reviewer Follow-up:** No response



### Reviewer dTuz

- **Score:** Initial **4**, Final **4** (Concerns Addressed & Positive Feedback)
﻿
- **Initial Review:**
Highlighted the **"Novel Insight"** (position > semantics) as a compelling finding. Major concerns included: the method being "prefill-only" (unexplored in long generation), efficiency overhead due to pseudo-tokens ($O(n^2)$), unclear generalization to NoPE/Hybrid architectures (e.g., Jamba), and the counterintuitive result where compression outperforms Full KV.
﻿
- **Author Response:** Author Response: We clarified DapQ's support for dynamic decoding compression and provided  **TTFT and Throughput benchmarks** alongside a theoretical proof (overhead scales with **$1/n$**), confirming **negligible latency**. We analyzed **Jamba (Hybrid/NoPE)**, clarifying that DapQ specifically targets **RoPE-based models** . Finally, we attributed the "Compression > Full KV" result to a "Denoising Effect" and validated DapQ's robustness in **Reasoning Mode (CoT)**.

- **Reviewer Follow-up:** We have addressed all the concerns of the reviewer. The reviewer showed positive feedback.


**We are confident the paper meets the high standards of ICLR.**

Best regards,

The Authors

---

### Note · Authors · 2025-12-29

**Comment:**

I have read and agree with the venue's withdrawal policy on behalf of myself and my co-authors.

**Withdrawal Confirmation:**

I have read and agree with the venue's withdrawal policy on behalf of myself and my co-authors.